**Cite this article:** Ålund M *et al.* 2022 Sensory environment affects Icelandic threespine stickleback's anti-predator escape behaviour. *Proc. R. Soc. B* **289**: 20220044.

behaviour, evolution

sensory evolution, anti-predator behaviour, global environmental change, plasticity, visibility, stickleback

**Author for correspondence:**
Murielle Ålund
e-mail: murielle.alund@ebc.uu.se

# Sensory environment affects Icelandic threespine stickleback's anti-predator escape behaviour

Murielle Ålund[1,2,3], Brooke Harper[2], Sigurlaug Kjærnested[4], Julian E. Ohl[5], John G. Phillips[2,3,6], Jessica Sattler[7], Jared Thompson[2], Javier E. Varg[1,2], Sven Wargenau[8], Janette W. Boughman[2,3] and Jason Keagy[9]

[1]Department of Ecology and Genetics, Animal Ecology, Uppsala University, Uppsala, Sweden
[2]Department of Integrative Biology, and [3]BEACON Center for the Study of Evolution in Action, Michigan State University, East Lansing, MI, USA
[4]Department of Aquaculture & Fish Biology, Hólar University College, Sauðárkrókur, Iceland
[5]Faculty of Environment and Natural Resources, University of Iceland, Reykjavík, Iceland
[6]Department of Biological Sciences, University of Idaho, Moscow, ID, USA
[7]Department of Biology, Miami University, Oxford, OH, USA
[8]Institute of Cell Dynamics and Imaging, University of Münster, Münster, Germany
[9]Department of Ecosystem Science and Management, The Pennsylvania State University, University Park, PA, USA

MÅ, 0000-0003-2861-9721; JEV, 0000-0002-7895-4563; JWB, 0000-0002-5731-7389; JK, 0000-0003-1235-4837

Human-induced changes in climate and habitats push populations to adapt to novel environments, including new sensory conditions, such as reduced visibility. We studied how colonizing newly formed glacial lakes with turbidity-induced low-visibility affects anti-predator behaviour in Icelandic threespine sticklebacks. We tested nearly 400 fish from 15 populations and four habitat types varying in visibility and colonization history in their reaction to two predator cues (mechano-visual versus olfactory) in high versus low-visibility light treatments. Fish reacted differently to the cues and were affected by lighting environment, confirming that cue modality and light levels are important for predator detection and evasion. Fish from spring-fed lakes, especially from the highlands (likely more diverged from marine fish than lowland fish), reacted fastest to mechano-visual cues and were generally most active. Highland glacial fish showed strong responses to olfactory cues and, counter to predictions from the flexible stem hypothesis, the greatest plasticity in response to light levels. This study, leveraging natural, repeated invasions of novel sensory habitats, (i) illustrates rapid changes in anti-predator behaviour that follow due to adaptation, early life experience, or both, and (ii) suggests an additional role for behavioural plasticity enabling population persistence in the face of frequent changes in environmental conditions.

## 1. Introduction

Habitats are rapidly changing because of disturbances such as habitat degradation, eutrophication and climate change, causing many animal populations to be exposed to novel environmental conditions. The capacity to adapt rapidly is key for species persistence in the face of such change. Understanding the behavioural, morphological and genetic changes underlying adaptation to new habitats is thus essential to predicting the consequences of changing environments on worldwide biodiversity. Linking changes in ecology to trait evolution in nature is often impeded by a lack of natural replication of the conditions studied. A powerful solution is using multiple populations subjected to similar ecological changes to infer links between changes in the environment and the traits of interest.

The arctic is warming rapidly with consequent ever-increasing rates of glacial melting, often forming new lakes if these glaciers are on land. In Iceland, some lakes formed by glacial melting are less than 50 years old [1], and the formation of new glacial lakes is accelerating [2,3]. Glacial lakes constitute a novel environment for the animal communities colonizing them, notably due to their high turbidity and hence low visibility with rapid decreases in brightness over short changes in depth (figure 1). Inhabiting such waters comes with challenges absent in nearby spring-fed lakes, which are typically crystal clear, even in deep water. Low levels of visibility in glacial lakes may select for increased use of other senses to acquire information critical for feeding, mating and avoiding predators [4–6]. Such changes in relative use of different sensory modalities and associated behaviours [6–8] are likely key processes facilitating adaptation to these novel environments [4]. Trait shifts could be caused by genetic evolution, plasticity or a combination of the two. The role of plasticity in favouring or hindering adaptation is still debated in the literature [9], as well as whether plasticity can be maintained after adapting to novel environments, or is instead lost through genetic assimilation of the favoured trait ('flexible stem' hypothesis [10,11]).

Here we seek to understand how colonization of highly turbid versus clear lakes affects anti-predator responses, focusing on differential use of senses. We use threespine stickleback fish (*Gasterosteus aculeatus*), considered an evolutionary model due to the exceptional capacity of marine populations to repeatedly invade freshwater habitats across the Northern Hemisphere [12,13], causing well-documented adaptation in physiology [12], morphology [14], life history and behaviour [12,13,15]. Iceland provides a unique opportunity to study rapid adaptation to novel environments because of a large network of geologically young (max. 14 000 to min. 50 years old) lakes spread across multiple watersheds, providing natural replication. In addition, differences in elevation affect the ease and timing of colonization. All freshwater river and lake populations are descended from marine fish, but populations at higher elevation (i.e. greater than 400 m above sea level, hereafter highland) are more isolated and likely more genetically distant from marine fish than those at lower elevation (lowland) because marine fish can regularly re-invade lowland lakes when their banks are breached by storms. This system thus represents a rare opportunity to determine the direction, magnitude and rate of evolutionary change by comparing multiple pairs of natural populations differing in both sensory ecology and isolation from ancestral marine fish.

We use the context of predator evasion to study how different visual conditions affect sensory cue use. Predation is a strong selective pressure for wild fish and typically stickleback rely heavily on vision to detect and escape predatory fish [16], primarily brown trout (*Salmo trutta*) and arctic char (*Salvelinus alpinus*) in Iceland. We expect that while stickleback from high-visibility, spring-fed lakes will still rely on vision for predator detection, adaptation to glacial lakes with low levels of visibility will favour reliance on other senses [17]. We focused on two important cues fish use to avoid predation by piscivorous fish: olfactory and mechano-visual [18]. While visual cues give accurate information on the nature and location of danger in high-visibility environments [6], fish can react very quickly to water movements caused by predators in close proximity via mechanoreceptors of their lateral line system [18]. Additionally, fish react to odours emitted by predators and injured conspecifics [19]. These olfactory cues can be detected at long range but lack precision, as they diffuse and persist in the water [7]. Visual, mechanosensory and olfactory cues can all elicit fear-like reactions, including erratic swimming behaviour, fast starts and freezing [18–20]. In nature, most fish presumably use a combination of sensory cues to best assess predation risk [4], but relative reliance on different modalities is likely affected by the sensory environment [21].

We experimentally measured anti-predatory responses of wild-caught sticklebacks from marine, lowland spring-fed, highland spring-fed and highland glacial lakes in laboratory trials, sequentially exposing fish to both olfactory and mechano-visual predator cues in a counterbalanced design. All populations were tested in two light conditions, simulating the visual properties of typical spring-fed and glacial lakes, as measured via spectrophotometry from representative lakes (figure 1), and thus representing both their native and a novel visual environment. Comparing reactions between light treatments allows us to infer the level of plasticity in anti-predatory behaviour of fish from different habitat types depending on immediate visual conditions. Our design further allows inferences on evolutionary changes in behaviour, by comparing the magnitude and direction of change between marine fish, which represent the ancestral state, and highland freshwater fish, expected to be most genetically distant from marine fish.

We predict that (i) fish differ in their overall reaction to predators depending on their native habitat, with the biggest differences expected between spring-fed and glacial habitats, as compared to putatively ancestral marine populations. This is because we expect that fish in spring-fed and glacial habitats evolved in different directions from the marine ancestor as they adapted to environments with very different visibility (electronic supplementary material, figure S1). (ii) Fish differ in their response to olfactory versus mechano-visual cues and the magnitude of this difference in response depends on their native sensory environment, with glacial fish relying more on olfaction than fish from spring-fed lakes. (iii) Fish from highland glacial lakes show either greater plasticity depending on immediate light conditions (if they retained ancestral predator response in high visibility), or show the least plasticity of all (if plasticity is lost through genetic assimilation). (iv) Differences between marine and highland populations are larger than between marine and lowland populations, due to longer time in isolation at higher elevation.

Below we present measures of anti-predator escape reactions from nearly 400 individuals from 15 populations and discuss how adaptive evolution and phenotypic plasticity may shape these behaviours in different habitats. Because of the recent time frame of population divergence and the fact that similar differences in visibility are caused by disturbances such as algal blooms or water pollution [22], our conclusions are relevant to a broad understanding of the consequences of rapid changes in sensory environments, and the interplay between plasticity and evolution in shaping population persistence.

## 2. Methods

### (a) Sampling procedures

Wild threespine sticklebacks (*Gasterosteus aculeatus*) were caught from 15 waterbodies in Iceland using minnow traps during

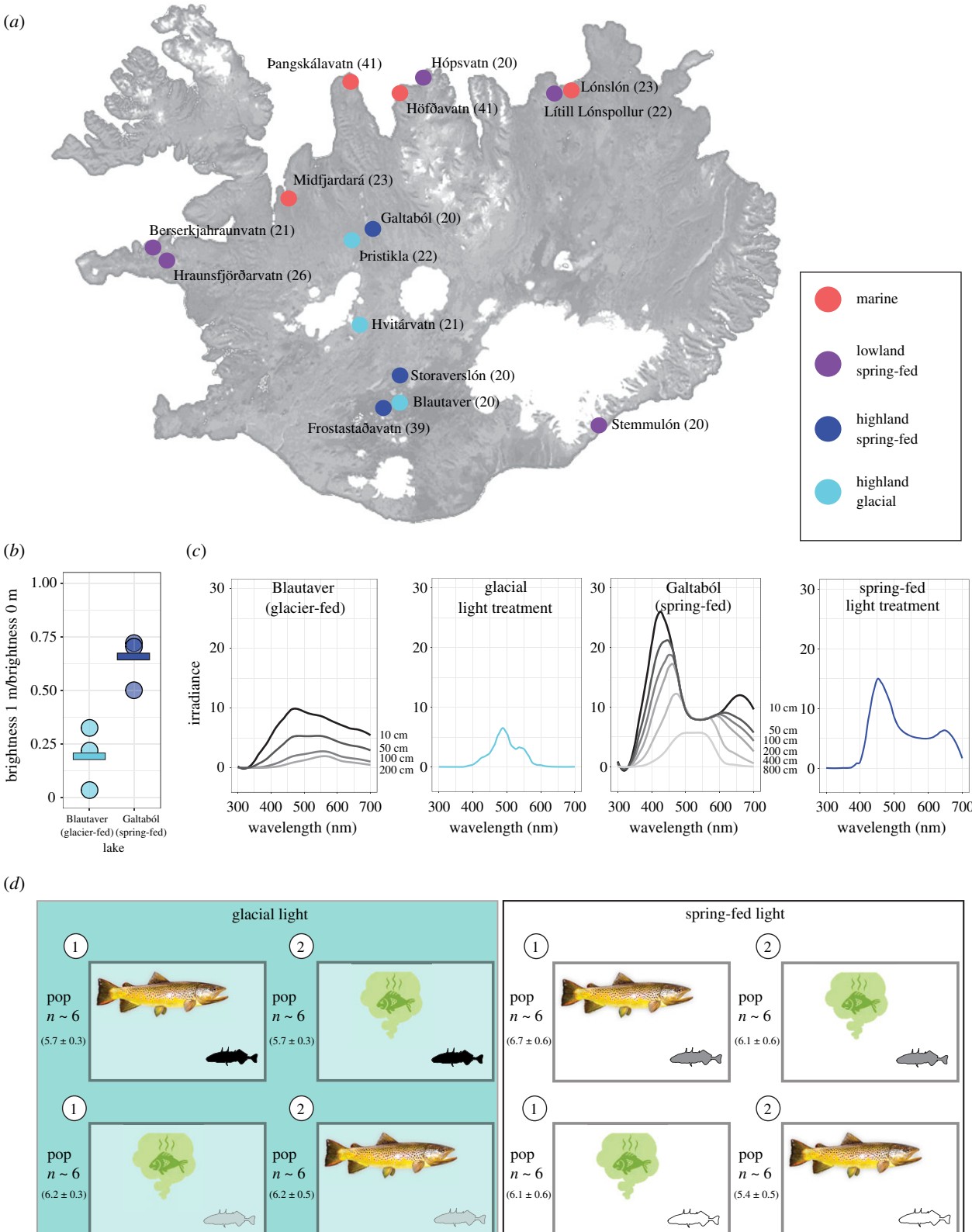

**Figure 1.** (a) Map of Iceland with glaciers in white and sampling locations colour coded by habitat type. The number of fish tested from each population are in parentheses. (b) Spectrophotometry measurements of a typical glacial and a typical spring-fed lake indicated very different proportions of surface light reaching 1 m depth ($t_{3.88} = -4.07$, $p = 0.016$). Plotted here are three sampling locations/lake (circles), with lake means (horizontal lines). (c) Irradiance depth profiles as measured in these two lakes and the corresponding light treatments for 2018 experimental tanks. Black spectra correspond to surface readings in lakes with decreasing greyscale corresponding to increasingly greater sampling depths indicated to the right of each curve. Irradiance values are directly comparable between light treatments because they are absolute (reported in photons per second and standardized to a light source) and in similar testing environments (no other light is available except that from the light used for the experimental treatment). Light spectra of the lakes could be impacted by weather or differences in sunlight angle. (d) Experimental design: fish were tested in one of two light treatments and sequentially encountered two predator cues (mechano-visual versus olfactory) in one of two orders (indicated by numbers in circles). Approximate sample sizes for each population for each light treatment × cue type × cue order combination are listed on the left, with mean ± s.e. in parentheses (see electronic supplementary material table S2 for additional details). The sample sizes for a given habitat type × light treatment × cue type combination are 27–63 (mean ± s.e.: 45.2 ± 3.3). (Online version in colour.)

summer 2017 and 2018 (figure 1). For 14 of these populations, irradiance was measured using a spectrophotometer to characterize brightness depth profiles (at 4–8 locations/lake and up to seven standardized depths from surface to bottom). Because ambient light conditions varied depending on time of day and weather, we used the proportion of light at the surface that made it to 1 m depth as a biologically relevant measure of brightness that was comparable between water bodies (electronic supplementary material, figure S1 and table S1). We sampled four marine sites (M, elevation 0–5 m, mean 1 m light transmission 47%), five lowland spring-fed lakes (LS, elevation 0–13 m, mean 1 m light transmission 41%), three highland spring-fed lakes (HS, elevation 457–593 m, mean 1 m light transmission 53%) and three highland glacial lakes (HG, elevation 420–574 m, mean 1 m light transmission 22%) (figure 1). Water bodies differed in light transmission ($\chi_3^2 = 10.39$, $p = 0.016$), which was entirely due to glacial lakes having much reduced light compared to other habitat types (estimate = $0.26 \pm 0.08$, $t_{3.13} = 6.04$, $p = 0.020$). Healthy, non-gravid adult fish were transported to Hólar University's aquatic laboratory (Verið in Sauðárkrókur, a lowland coastal town) and housed 2–4 weeks in 13 l opaque mixed-sex tanks by population, with continuous water flow, at the temperature and salinity measured for each population (4–14°C, salinity: freshwater: 0–1 ppt, marine: 31 ppt). Fish were fed daily *ad libitum* with frozen bloodworms.

## (b) Behavioural trials

We simulated predatory attacks on 378 individuals (20–41 fish per population; figure 1 and electronic supplementary material, table S2 for detailed breakdown) in two plexiglass tanks with dark walls (123 × 38 × 30 cm, filled to 21 cm), containing two plastic refugia at opposite ends and a grid of 12 rectangles of 20 × 9 cm drawn on the bottom (electronic supplementary material, figure S2). Each fish was selected haphazardly from its holding tank (gravid females were avoided) and was sequentially subjected to both predator cues in pseudo-random order, in one of two light treatments, with the aim to test equal numbers of fish in each light treatment for each population (figure 1). Experimentally manipulating suspension of solids to simulate turbidity is challenging and instead we focus on the reduction in brightness that turbidity induces, which is expected to have strong effects on animals such as stickleback that are strongly dependent on vision (figure 1). Our light treatments primarily involved manipulating brightness to simulate levels measured in either a representative spring-fed (full spectrum fluorescent bulb) or glacial lake (full spectrum bulb wrapped in filters, figure 1b,c; electronic supplementary material, methods). Our manipulation resulted in the glacial light treatment having at most 26% of the light of the spring-fed light treatment, which is an equivalent reduction in light as the one experienced by a fish in a spring-fed lake that swims from the surface to 6 m depth (calculations based on figure 1b,c). The experimental tanks were separated by black-out curtains and light treatment exchanged between tanks half-way through the experiment each year.

Two types of cues simulated an attack by a piscivorous salmonid fish: (i) mechano-visual cue: a 250 mm rubber trout was propelled through the tank at a speed of $0.77\ \mathrm{m\ s}^{-1}$, activated by pushing a button (electronic supplementary material, Methods), simultaneously giving the stickleback subject visual and mechanical stimulation (water movement). The model predator was covered with opaque plastic during the olfactory trial. (ii) Olfactory cue: alarm cues, chemical substances released from the skin of fish in response to mechanical damage, were obtained from stickleback from the same population as each subject fish. In 2018, to increase the response, concentrated odour from arctic charr (predator cue) was added to the alarm cues (electronic supplementary material, Methods). Olfactory cues

were released in the experimental tank by pushing a button activating a pump releasing 40 ml of liquid at $0.8$–$1.2\ \mathrm{l\ min}^{-1}$.

Before trials with a new fish, experimental tanks were cleaned and filled with water matching temperature and salinity of the subject's original habitat. All trials were recorded from above using a Canon Vixia HF R72 camcorder and simultaneously projected on a monitor behind a black curtain for live scoring. After an acclimation period of equal length for all fish, the first predator cue was released when the subject fish entered the centre of the experimental arena, either facing the robotic predator, or in proximity to the hose delivering the olfactory cue. Initial reaction was recorded for 2 min, followed by 8 min of activity scoring (recording each gridline cross, see electronic supplementary material, Methods). Ten minutes after the release of the first cue, and after another acclimation period, the second cue was presented and initial reaction and activity scored again. The trial was aborted if the subject did not reach the correct position to trigger the cue. At the end of a full trial, fish length and weight were measured, sex was confirmed by examining gonads and the presence of parasites in the body cavity noted.

Reaction distance to mechano-visual cues was measured from recordings using ImageJ for 280 fish from 13 populations (some excluded due to technical issues). We measured the nose-to-nose distance between the robotic predator and the stickleback when it showed the first reaction to the attack (electronic supplementary material, Methods).

## (c) Statistical analyses

Statistical analyses were performed using R (v. 4.0.2 [23]) and the packages 'lme4' and 'car' for generalized linear mixed models, 'emmeans' for contrasts and 'ggplot2' and 'beanplot' for graphs. Below we report results of analysis of deviance with type III Wald tests (with associated $\chi^2$ statistics); linear model coefficients are in electronic supplementary material, tables.

In four generalized linear mixed models, we tested the effect of habitat type, light treatment and predator cue on the given response variable (initial reaction fast start, initial reaction freeze, reaction distance or activity level), with the additional fixed effects of year (2017 or 2018) and trial order (first or second trial for the fish), and the random effects of individual fish (accounting for repeated measures), population and observer. Interactions between habitat type, light treatment and predator cue were included in initial models and removed stepwise when not significant. Contrasts were performed to understand significant interactions and main effects; *p*-values were corrected by the corresponding number of tests on the same dataset. Note that neither sex nor parasitism significantly affected the results relevant to our predictions (electronic supplementary material, table S7). These variables were thus not included in the final models. Fish were more active but reacted slower to the mechano-visual cue in the first compared to the second trial (trial order effect, electronic supplementary material, tables S4 and S5). Fish were more active, reacted slower to the mechano-visual cue, and were more likely to use fast start with olfactory cue in 2018 than 2017 (year effect, electronic supplementary material, tables S3–S6). We attribute between-year differences to improvements in experimental design between years, meant to increase participation of fish (electronic supplementary material, Methods); we control for the average differences in behaviour induced by these changes by including year as a fixed effect. We did not add the seven additional interaction terms required to fully specify the year effect as these models did not converge, but those interactions are not of specific interest for hypotheses tested in this study. Rather, the consistent effects emerging despite year-to-year variation are our focus.

We extracted semi-partial $R^2$ from our models following [24], although we refer readers to read that reference for caveats in interpreting these effect size estimates for models such as ours. See electronic supplementary material, Methods for more details and the full list of effect sizes in the electronic supplementary material, tables S8–S11. Marginal and conditional $R^2$ for all our full models are reported below, extracted using the r.squaredGLMM function in the MuMIn R package [25]. All data and code are available on Dryad [26].

## 3. Results

We first present interaction effects, or if there were none, the significant main effects, from analyses of deviance of our four main models. Then in subsections, we present relevant analyses testing our four major predictions.

Habitat type, light treatment and predator cue all significantly affected the likelihood of a fast start, with no significant interactions ($\chi^2_3 = 9.96$, $p = 0.019$, $\chi^2_3 = 4.91$, $p = 0.02$, $\chi^2_3 = 165.2$, $p < 0.001$, respectively, figure 2 and electronic supplementary material, tables S3, S8, S12 and S13, marginal $R^2$: 0.30, conditional $R^2$: 0.40). Light treatment and predator cue significantly affected the probability of freezing, with no significant interactions ($\chi^2 = 7.90$, $p = 0.005$ and $\chi^2 = 31.98$, $p < 0.001$, respectively, figure 2, electronic supplementary material, tables S3, S12 and S13, marginal $R^2$: 0.41, conditional $R^2$: 0.45). The distance at which fish reacted to the mechano-visual predator cue was significantly affected by habitat type ($\chi^2_3 = 10.18$, $p = 0.017$, figure 3; electronic supplementary material, table S4), with no effect of light treatment ($\chi^2 = 1.42$, $p = 0.234$, marginal $R^2$: 0.09, conditional $R^2$: 0.10) or significant interactions. Finally, there was a significant three-way interaction between habitat type, predator cue and light treatment on activity levels ($\chi^2_3 = 42.4$, $p < 0.001$, figure 4, electronic supplementary material, table S5, marginal $R^2$: 0.31, conditional $R^2$: 0.98) that we dissect further below.

### (a) Does habitat type affect anti-predator behaviour?

Habitat type significantly affected the probability to react with a fast start ($\chi^2_3 = 9.96$, $p = 0.019$). Highland spring-fed fish were less likely than marine fish to use a fast start regardless of cue or light treatment ($z = -2.96$, $p = 0.003$, figure 2; electronic supplementary material, table S3); the other habitats did not differ from marine fish.

Fish also differed in their reaction time depending on habitat type ($\chi^2_3 = 10.18$, $p = 0.017$, figure 3; electronic supplementary material, table S4). Fish from spring-fed lakes reacted fastest to a predator (longer distance from predator at first reaction; lowland spring-fed: $15.62 \pm 4.36$ cm (mean ± s.d.), highland spring-fed: $15.27 \pm 5.24$ cm, whereas fish from highland glacial lakes were slowest to react ($12.66 \pm 6.66$ cm), as might be expected if their visual systems are less effective.

Habitat type, cue and light treatment all interacted to cause differences in activity (figure 4, more below). Overall, activity was highest for fish from highland spring-fed lakes (contrast comparing highland spring-fed to all other types: estimate $= 1.23 \pm 0.61$, $z = 2.00$, $p = 0.045$; line crosses in 8 min: highland glacial: $32.2 \pm 3.2$, lowland spring-fed: $37.9 \pm 2.1$, highland spring-fed: $48.8 \pm 3.0$, marine: $38.9 \pm 3.0$).

### (b) Are there differential responses to olfactory or mechano-visual cues, and does this vary with habitat type?

Sticklebacks had different initial reactions to the two types of predator cues. Fish were more likely to use fast start in reaction to the mechano-visual than the olfactory cue (226/361 versus 45/362 occurrences, $z = 12.85$, $p < 0.001$, figure 2a) and were instead more likely to freeze in reaction to the olfactory cue (98/362 occurrences, $z = -5.66$, $p < 0.001$, figure 2a; electronic supplementary material, table S3).

Activity was significantly lower following exposure to the mechano-visual than the olfactory cue ($37 \pm 2$ and $41 \pm 2$ line crosses, respectively, contrast estimate $= 0.605 \pm 0.1$, $p < 0.001$, figure 4a,b). To test for differential use of sensory systems due to adaptation to different environmental conditions, we compared fish from the two highland habitat types, likely most diverged from ancestral marine sticklebacks, in the light treatment that simulated conditions of their respective native environment. Highland glacial fish showed strongly reduced activity after the olfactory compared to the mechano-visual cue in the glacial light treatment, while highland spring-fed fish in spring-fed light showed the opposite pattern (figure 4c); these opposite reaction norms were significantly different from one another (contrast estimate $= 0.37 \pm 0.05$, $z = 6.95$, $p < 0.001$).

### (c) Is there plasticity in anti-predator response depending on immediate visual conditions, and does this vary between habitat types?

Light treatment affected initial response to predation for all fish, independent of habitat type and cue: fish were more likely to escape with a fast start in spring-fed than glacial light ($z = 2.21$, $p = 0.027$) and more likely to freeze in glacial than spring-fed light treatments ($z = -2.81$, $p = 0.005$).

To test for plasticity in activity depending on current visual environment, we computed contrasts between light treatments for each habitat type and predator cue. Highland glacial fish were the only ones showing plasticity between sensory environments, with significantly higher activity in glacial than spring-fed light after the mechano-visual cue (contrast estimate $= -0.50 \pm 0.22$, $z = -2.24$, $p = 0.025$, figure 4d, electronic supplementary material, table S6 for all contrasts).

### (d) Does putative divergence time impact the magnitude and direction of behavioural changes?

Here we compare high- and low-elevation spring-fed lakes to marine populations, with elevation acting as a proxy for genetic isolation. If adaptation to the spring-fed environment is occurring, we expect the difference between highland spring-fed and marine fish to be larger than, but in the same direction as that between lowland spring-fed and marine populations.

In support of this prediction, highland fish showed a significant reduction in use of fast start compared to marine fish (estimate: $-4.33 \pm 1.46$), but lowland fish did not significantly differ from marine fish in use of fast start ($-0.75 \pm 1.41$), (contrast: $-3.58 \pm 1.47$, $z = -2.43$, $p = 0.015$). Highland spring-fed fish were also more different from marine fish than were lowland fish in their probability to freeze ($3.93 \pm 1.68$, $z = 2.34$,

(a)

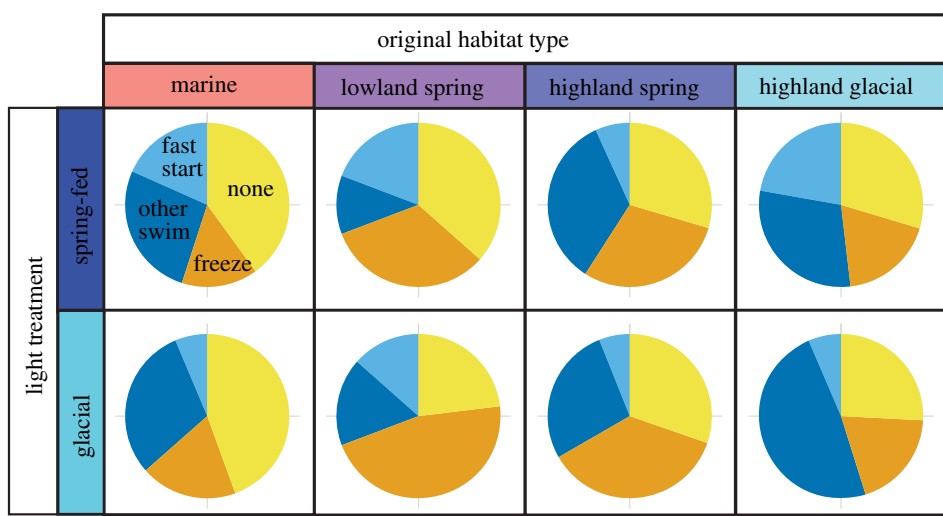

(b)

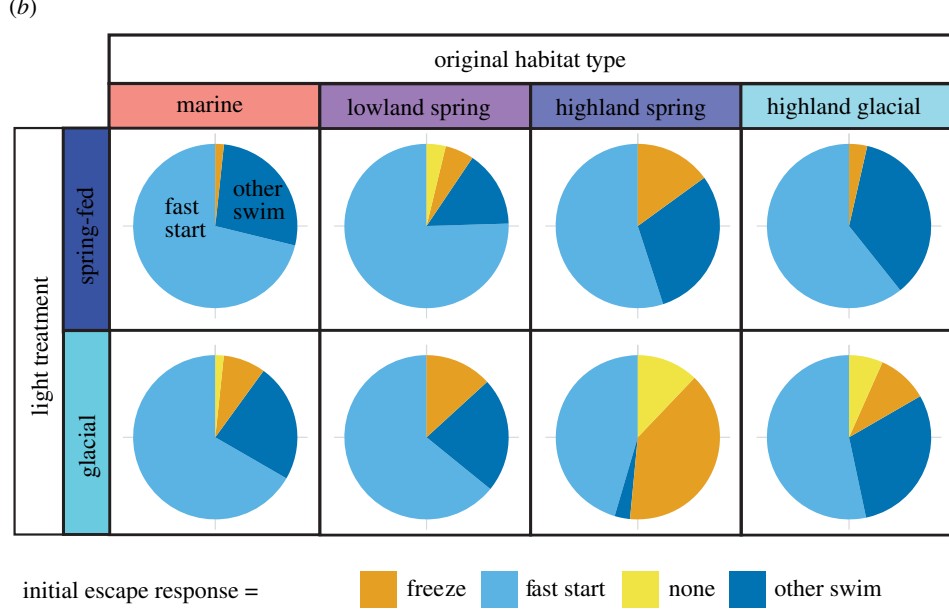

initial escape response =    freeze    fast start    none    other swim

**Figure 2.** Initial escape reactions to (a) olfactory and (b) mechano-visual predation cues. The proportion of fish using one of four initial reactions to each of the predatory cues (fast start, freeze, other swim and none) are presented by original habitat type (marine, lowland spring-fed, highland spring-fed and highland glacial, columns) and experimental light condition (glacial and spring-fed light, rows). The total number of fish using each reaction in all combinations of habitat type and light treatments are presented in the electronic supplementary material, tables S12 and S13. (Online version in colour.)

$p = 0.019$), with highland fish more likely to freeze (3.81 ± 1.92), while lowland fish were indistinguishable from marine fish (−0.12 ± 1.8). By contrast, activity was significantly reduced in lowland spring-fed fish (−1.01 ± 0.68), but not different in highland fish compared to marine fish (0.794 ± 0.705), thus making lowland spring-fed fish more different for this behaviour (1.81 ± 0.72, $z = 2.5$, $p = 0.012$).

## 4. Discussion

We show that habitat of origin, predator cue and light environment all affect threespine stickleback anti-predator behaviour, often in interacting ways. The influence of habitat type suggests that divergent selection pressures led to changes in the use of sensory modalities and anti-predator behaviour. The interaction between predator cue, habitat type and light treatment on activity further suggests populations from different habitats vary in their behavioural

plasticity. Habitat-associated changes in behaviour could reflect evolutionary change, but also could involve learned responses or result from other forms of developmental plasticity. Highland and lowland spring-fed lakes differ in their expected genetic divergence from marine populations and comparing them could give insights into possible genetic evolution. However, further experimental and genetic work, ideally involving reciprocal transplants between habitat types, is needed to parse out the relative role of genetic differences, developmental plasticity and interactions between the two in determining the habitat-associated behavioural differences we report here. We discuss our findings more fully below.

Fish activity was lower after being exposed to mechano-visual compared to olfactory cues, especially in high-visibility conditions. Furthermore, all fish were more likely to react to olfactory cues with fast start in spring-fed than glacial light, and were more likely to freeze in glacial than spring-fed light. Low activity is thought to indicate high

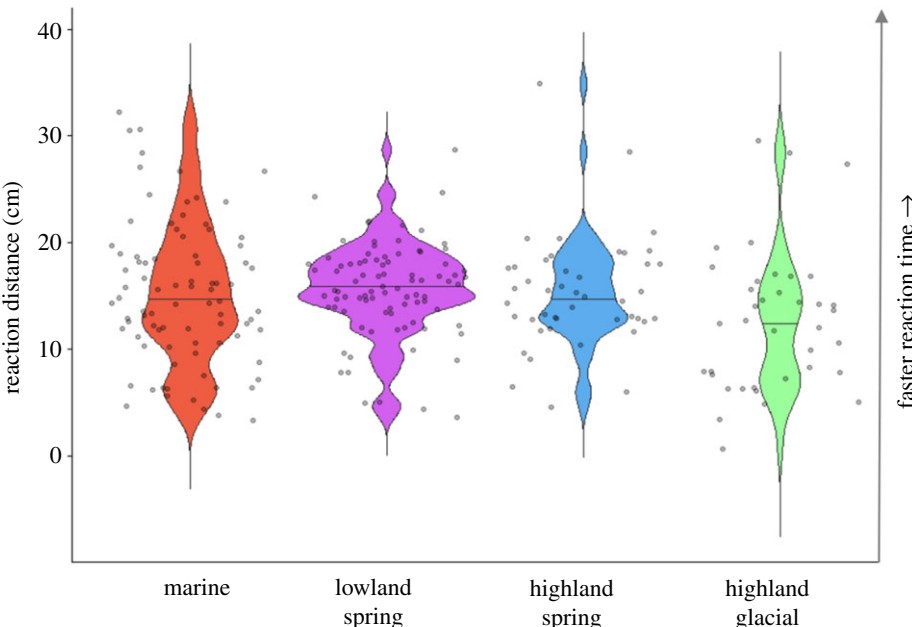

**Figure 3.** Distance between the stickleback and the robotic predator when the first reaction was observed. A shorter distance means that the fish took longer to react to the simulated predation event. Violin plots show the frequency of each distance value (wider means more frequent), with the median value for each habitat depicted with a horizontal bar. Individual data points are indicated as filled circles. (Online version in colour.)

levels of fear [27,28], translating into longer time to resume exploring, likely reducing detection by predators. The less drastic response to olfactory cues generally, and to both cues in low-visibility conditions, presumably reflects poor information on both the position and timing of a threat, and reduced ability to determine a safe escape route or refuge location. Such a lack of information could be deadly when visual conditions are impaired [6,7].

Fish from both highland lake types stood out as being different from other habitats. Highland spring-fed fish were least likely to use fast start, had the fastest reaction times and were most active overall. Highland glacial fish in contrast had the slowest reaction times and were alone in reducing activity when exposed to olfactory as opposed to mechano-visual cues, particularly when visibility was low, thus reacting most strongly to olfaction. Highland lakes are different from lowland and marine habitats in several ways, including greater seasonality and harsher conditions, which represent strong selective pressures. And yet, highland spring-fed fish differed substantially in behaviour from highland glacial fish despite being geographically proximate, suggesting that these shared aspects of the environment were not driving the differences we saw, but rather that the different visual environments are key. The high-visibility conditions of highland spring-fed lakes likely favour increased visual acuity and/or reliance on vision, allowing these fish to detect danger quicker (the long reaction distances we measured make it unlikely mechano-sensation was used in these trials), and be better able to judge when energetically expensive escape behaviour is necessary, for example by limiting their use of fast start. In glacial lakes, the sharp decline in brightness as depth increases (figure 1) may render vision unreliable and favour increased reliance on olfaction and mechano-sensation for short-range reactions instead. Similar shifts away from vision in low-light environments are seen in other organisms [5,6,29–32]. When we compared highland fish reactions in their simulated native visual environments,

we saw a strong divergence in preferred sensory modality further suggesting a match between native visual environment and sensory cue use expected with local adaptation or developmental plasticity. Given how young Icelandic highland lakes are, the differences between the highland lake habitats and the lowland and marine habitats must have occurred very rapidly. Future experiments using common-garden experiments will be necessary to disentangle any potential effect of early life experience from genetic differences.

While our current study design makes it difficult to tease apart the relative influence of local adaptation and developmental plasticity, we can make some inferences about the role of genetic divergence on behavioural changes due to how geography constrains gene flow in Icelandic stickleback fish. Specifically, highland lakes were formed after glacial retreat and then colonized by marine sticklebacks working their way upstream. Because stream reaches are long and flow can be substantial, many generations are required to reach and colonize highland lakes; their populations thus remain isolated from marine populations. By contrast, many of the lowland lakes are close to the sea and sometimes reconnect to the ocean, allowing gene flow between lowland and marine populations. Thus, we expected highland fish to differ most from other populations, which was apparent in most analyses. Focusing on comparisons between highland and lowland spring-fed populations allows us to test for effects of presumed divergence time on changes in antipredator behaviour, independent of the effect of sensory environment (because both have similar visibility, electronic supplementary material, figure S1). These analyses support a reduction in fast start and increase in freezing behaviour in putatively derived highland populations. Frequent fast start behaviour is associated with high levels of perceived predation risk and is energetically costly both directly (reduced foraging time) and indirectly (high stress) [32,33]. The harsh conditions experienced in the highlands, with a

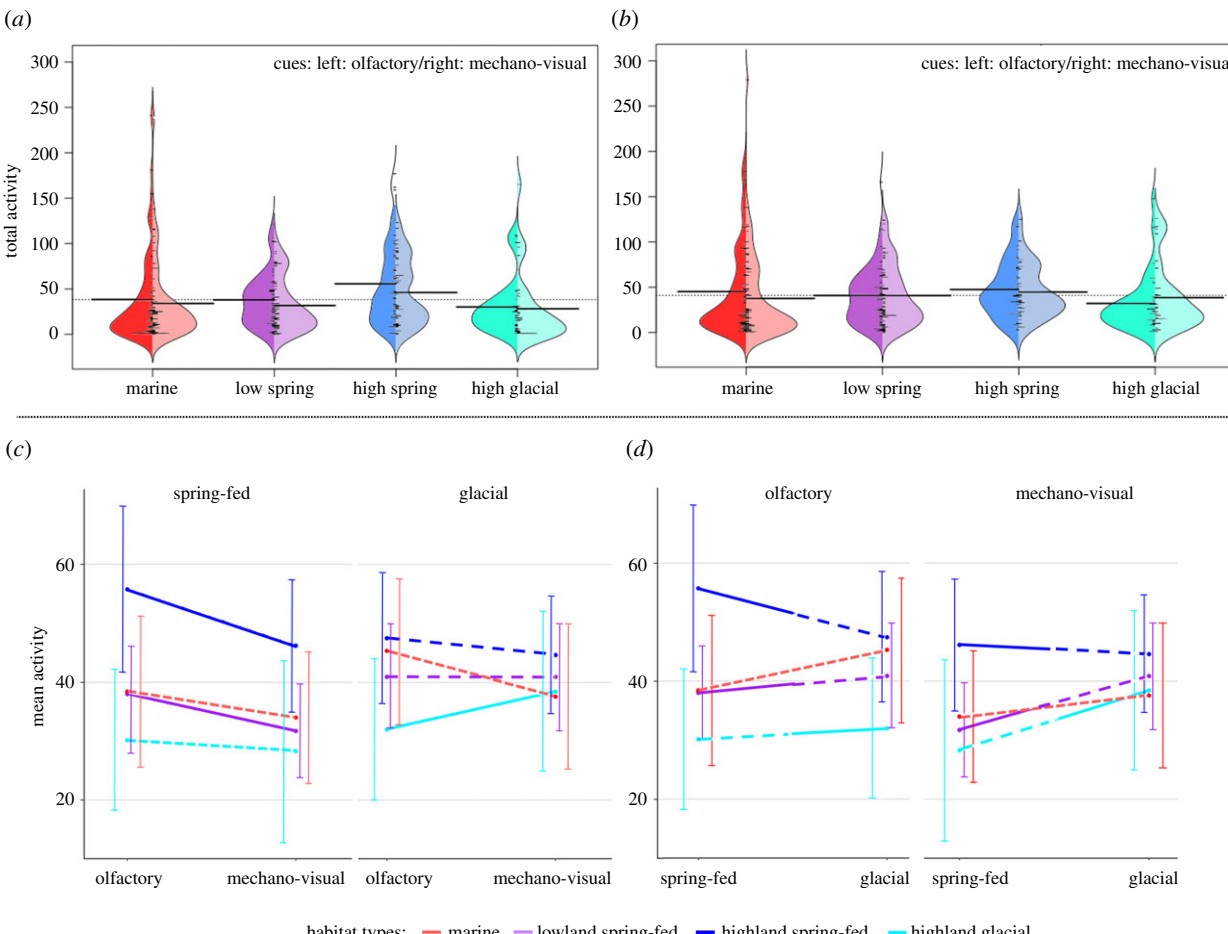

**Figure 4.** Fish activity over an 8 min observation period after being exposed to either the olfactory or mechano-visual predator cue, in glacial or spring-fed light. Activity scores are grouped and colour coded by original habitat type (marine, lowland spring-fed, highland spring-fed and highland glacial). (*a,b*) Bean plots representing the entire dataset where activity after olfactory cue is depicted on the left side of each individual 'bean' and activity after the mechano-visual cue is on the right side, in spring-fed (*a*) and glacial (*b*) light. Long horizontal solid black lines represent mean values for each subset of data, while the dotted lines across each panel represent mean activity across each respective experimental light treatment. Within each half of each 'bean' is a histogram of the relevant data. (*c,d*) Reaction norms depicting mean activity and 95% confidence intervals, comparing reactions to the two different cues in each respective light environment (*c*) and reactions to the two different experimental light environments for each respective predator cue (*d*). The solid lines represent trials where fish were tested in their 'native' light environment (glacial light for highland glacial fish, spring-fed light for lowland and highland spring-fed fish), and the dashed lines represent trials in a novel light environment. Habitat types from top to bottom (*c* and *d*): high spring, marine, low spring, high glacial. (Online version in colour.)

very short period where lakes are not frozen, likely select for very fine-tuned and less energy-demanding anti-predator behaviour in these populations.

Different reactions to the same predator cue between light conditions point to population-level variation in plasticity for anti-predator behaviour. Interestingly, highland glacial fish differed most in activity between light environments in response to the mechano-visual cue, with relatively higher activity in low visibility compared to high visibility. Predators are presumably also constrained by reduced visibility and less able to detect prey from afar [6]. A quicker return to normal activity after signs of danger have disappeared should be advantageous, given that time spent performing anti-predator behaviour trades-off against time foraging or reproducing [34]. Glacial lakes not only have poor visibility relative to spring-fed lakes, but also much steeper gradients in light intensity as fish descend (figure 1). Plastic responses to light conditions may thus be adaptive in these lakes, because fish are exposed to very different light levels over small depth differences. In other species, sensory systems show plasticity in response to sensory environment [35] as does anti-predator behaviour [36], and plastic anti-predator responses are thought to underlie some

coral reef fish's ability to persist in the face of anthropogenic sediment influx [33]. Paired with the finding that glacial fish also rely more on olfaction than other fish in their native light environment, this suggests, perhaps surprisingly, that both local adaptation and increased behavioural plasticity has occurred. These forces are often expected to trade off against each other [11], but see [9] for a recent report in *Daphnia* of high variation in ancestral plasticity that potentially favours adaptation to novel predators and is maintained in derived populations. Finding that plasticity is increased in derived glacial populations refutes the flexible stem hypothesis and instead suggests a combination of increased plasticity and local adaptation might be key to successful colonization and persistence in drastically different habitats.

Rapid anthropogenic change can influence behavioural adaptation and fitness, and aquatic environments are disproportionally affected by climate change, which is further accelerated at high latitudes [3,22,37]. Behavioural interactions between species are especially sensitive to changes in sensory environments such as increased turbidity [6] and corresponding changes in behaviour have been documented in multiple aquatic organisms [5,6,8,17,29,33,38]. Quantifying

the pace of behavioural evolution and the relative contribution of local adaptation and plasticity is, however, still challenging in wild organisms. The results of our study, taking advantage of a unique geographical setting with multiple populations differing both in the degree of isolation and environmental pressure, goes one step forward towards a more global understanding of the complexity and repeatability of behavioural change in novel environments. Genetic analyses will allow definite conclusions about the speed of evolution, but the short time available for colonizing Icelandic freshwater lakes since the last glaciation allows us to conclude that anti-predator responses have changed rapidly, and most strongly in more isolated populations. Our experimental design further suggests that a combination of adaptation and enhanced behavioural plasticity allowed sticklebacks to colonize and persist in the most novel visual environments. We encourage additional research to further our understanding of the potential of other taxa to show both local adaptation and behavioural plasticity in fitness-related traits. Because sensory systems are necessary for communication between animals and assessment of their surroundings essential for survival, and adaptation to new sensory environments may involve many changes in development, morphology, behaviour and neurological pathways, understanding the consequences of habitat disturbance on sensory adaptation is of central importance for predicting the impact of habitat change on biodiversity at large [39].

Ethics. All experimental procedures were realized in accordance with regulations and ethical approval by the Michigan State University Institutional Animal Care and Use Committee (IACUC, protocol numbers 05/18-077-99, 05/16-064-00 and 201900128) and the Icelandic Food and Veterinary Authority (Matvaelastofnum, MAST). Permits to collect stickleback fish were granted by Fjállabak Nature Reserve and the Vantajökulsþjóðgarður National Park.

Data accessibility. The dataset and code supporting this article are available from the Dryad Digital Repository: https://doi.org/10.5061/dryad.70rxwdc0b [26]. The data are provided in the electronic supplementary material [40].

Authors' contributions. M.Å.: conceptualization, data curation, formal analysis, investigation, methodology, supervision, validation, visualization, writing—original draft and writing—review and editing; B.H.: investigation and methodology; S.K.: investigation; J.E.O.: investigation; J.G.P.: conceptualization, investigation and methodology; J.S.: investigation and methodology; J.T.: conceptualization, investigation, methodology and resources; J.E.V.: investigation; S.W.: data curation, investigation, methodology and software; J.W.B.: conceptualization, funding acquisition, investigation, methodology, project administration, resources, supervision, validation, writing—original draft and writing—review and editing; J.K.: conceptualization, data curation, formal analysis, funding acquisition, investigation, methodology, project administration, resources, software, supervision, validation, visualization, writing—original draft and writing—review and editing.

All authors gave final approval for publication and agreed to be held accountable for the work performed therein.

Competing interests. The authors have no competing interests.

Funding. Funding for this project was provided by NSF Dimensions of Biodiversity grant (grant no. DEB-1638778) to J.W.B. with subaward to J.K. (AE671 SBC Michigan State RC106522UI), BEACON Center for the Study of Evolution in Action (NSF DBI-0939454) to J.W.B. and J.K., a Fulbright Fellowship in Arctic Research to J.W.B., the Swiss National Science Foundation (Early Postdoc Mobility Fellowship no. P2SKP3_184052) to M.Å. and Liljewalchs resestipendium and Stiftelsen för Zoologisk Forskning to J.E.V. J.K. was also supported by the USDA National Institute of Food and Agriculture Federal Appropriations under project PEN04768 and accession number 1026660.

Acknowledgements. The authors would like to thank Bjarni Kristófer Kristjánsson, Kári H. Árnason, Skúli Skúlason, Rakel Þorbjörnsdóttir, Greg Byford and Brielle Dominguez for their support with field work and experimental procedures in Iceland. We thank all landowners in Iceland that generously allowed us access to lakes for sampling. We thank two anonymous reviewers and an Associate Editor for their input that greatly improved this manuscript.

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
