## [Peer Review File · Proceedings of the Royal Society B: Biological Sciences]

Review History

RSPB-2021-0022.R0 (Original submission)

Review form: Reviewer 1

Recommendation

Major revision is needed (please make suggestions in comments)

Scientific importance: Is the manuscript an original and important contribution to its field?

Good

General interest: Is the paper of sufficient general interest?

Good

Quality of the paper: Is the overall quality of the paper suitable?

Marginal

Is the length of the paper justified?

No

Should the paper be seen by a specialist statistical reviewer?

No

Do you have any concerns about statistical analyses in this paper? If so, please specify them explicitly in your report.

Yes

It is a condition of publication that authors make their supporting data, code and materials available - either as supplementary material or hosted in an external repository. Please rate, if applicable, the supporting data on the following criteria.

Is it accessible?

Yes

Is it clear?

Yes

Is it adequate?

Yes

Do you have any ethical concerns with this paper?

No

Comments to the Author

This study sought to investigate behavioural adaptations of Icelandic sticklebacks populations to the sensory environment of with different habitats. The authors collected a large number of sticklebacks from the wild from 15 different populations and tested them individually in the lab on their anti-predator behaviour. Fish were tested in two light conditions meant to resemble the clear and turbid waters of marine water/spring-fed lakes and glacial lakes respectively and both with a moving model predator and an olfactory cue. The authors found that habitat type, the light treatment, and predator cue all influenced the fish's anti-predator behaviour, often in complex ways.

I'd like to start by stating the authors did an impressive amount of work, testing close to 400 individual fish from many different populations and habitats. I also think there is potentially great value in the dataset the authors collected to help better understand behavioural adaptations. I have however a number of issues with the paper that warrant me from recommending it for publication.

First, the authors frame their paper in the context of human-induced environmental change but actually did not study behavioural responses to such change but to different natural habitats. It is fine to have a couple sentences in the discussion to state the potential relevance for such work, but framing the study in this way is not appropriate. It is also not needed as it is interesting enough to understand how the populations from the different natural habitats diverged from one another

Second, one of the main focus points of the paper seems to be the difference in turbidity between the habitat types. However, rather than testing fish in different levels of turbidity they tested fish in two different light conditions. The authors do not provide any clear reasoning for doing so nor enough detail to understand how fish perceive the light conditions or tests if the 'glacial light' condition indeed resembles turbidity by providing less visibility. It is also not considered that besides a reduced visibility, the change in brightness and colour of the light may give the fish a different sense of safety and the behavioural responses observed may in that case be more to do with the risk-taking behaviour than with sensory input.

Third, the experimental design was overly complex, with fish from four different habitat types tested with two light types and two odour cues. The authors thereby ran models with complex interactions and did a poor job in explaining the results, making it very hard to extract the key findings.

Fourth, the paper was really lengthy and far from to the point with for example the introduction

being close to double the length of many ProcB papers. Also in general I found the authors did a poor job in explaining what they did and why, and in good order, such as stating experimental conditions before explaining the experimental design and very lengthy, hard to follow predictions without proper understanding of the experimental details. In combination with the complex interactions between the different factors and hard to follow results, the paper unfortunately does a poor job in explaining a potentially promising dataset.

Below I provide more detailed comments on the manuscript.

Lines 21-24: See my main point regarding the framing of the study in the context of environmental change.

Line 24: Why novel sensory environments? Fish moving to different habitats will experience different environmental conditions yes, but the environments are not necessarily novel (sensory speaking).

Line 28: Strictly speaking the population's habitat did not affect anti-predator behaviour but was linked to it. Or in other words, you found that fish's anti-predator behaviour could be explained by their habitat.

Lines 70-73: A more complex genetic architecture does not necessarily mean evolution is slower; it could also be faster.

Lines 77-78: This is not the place yet to state what was studied and should be moved down or better, explained in a different way.

Lines 79-81: This is important information that should come way earlier in the introduction. Actually, it would make more sense to me to write the paper around this focus rather than starting with the climate change narrative, which is not so fitting to the actual study.

Lines 86: What is meant with "current and historical" exactly? Make clear in the text.

Lines 86-87: If predation/predator evasion is suggested as a main potential selective pressure for sticklebacks in Iceland you should provide more detail about that here, such as the sympatry with predators and occurrence of which species. This is critical information currently missing from the manuscript (apart from a sentence in the discussion).

Lines 92-93: Again, information is provided about some part of the practical work before any real explanation of the study. Leave this for later in the introduction.

Line 100: burst swimming is not the same as a fast start and should not be confused with one another.

Lines 103-122: This is a paragraph of 288 words meant to explain the study and the predictions. This paragraph is really way too long and is hard to follow. Furthermore, despite its length, the study itself is actually still not well described (only 2 lines) and much remains unclear. For example, did you conduct an experimental or observational study? Was it done in the lab? What is meant with "light environments" and is this experimental design appropriate for investigating responses to turbidity? Etc. State concisely what you did exactly, with what purpose, and what predictions. The predictions do not need the long explanations given here when the proper background information is given above.

Lines 125-128: Here, the last three sentences of the introduction, the link is made again between turbidity and the potential relevance for human induced environmental change. This point is fine but is more for the end of the discussion, and framing the whole study in this context is not appropriate, see main point.

Line 133: You have to be more specific about where you caught the sticklebacks than "in Iceland". Especially as you are testing fish from different types of sites and multiple replicates per site. I later found some of this information is provided in the SM. Integrate this and actually refer to that information.

Line 143: Clarify that tests were done with individual fish. How were these focal fish selected? How was the order of testing decided? Was population sufficiently randomized across trials? Much detail is missing about the experimental design.

Lines 147 onwards: Not sure why some words are underlined, but not a styling that ProcB supports as far as I know.

Lines 148-151: With what speed? Was this controlled and consistent across all trial? Was the replica already visible at the start of the trial or how was this prevented?

Lines 157-161: Given that turbidity is one of your main interests and main focus points of the study it is unclear to my why you not use turbidity itself in your experimental design but rather

use a difference in “light environment” instead. Furthermore, you need to explain a lot more about these treatments, such as why you used light environments, what do the light environments entail, how are they perceived by the fish, and if pilots have been done to test if they are a good proxy for testing of turbidity. Did you run any tests to measure underwater visibility and was that lower for the ‘glacial light’ environment? Has such an experimental paradigm used before at all?

Lines 168-170: This is unclear. So each trial a fish received a second cue? Why was that done? Very likely the first cue will influence the second cue, making it much harder to disentangle the effects than were they tested separately. Also, was made sure the order of the cues was randomized?

Lines 175: What happened with the other 98 fish (~25% of the experimental fish)?

Line 185: So each fish received two trials? This is not clearly stated anywhere before, neither how much time there was between trials, if that was the same for all fish, etc.

Lines 186-187: You state interactions were included. But as you tested five fixed factors, what interactions did you actually include? And did you actually have some clear hypotheses for including all interactions?

Results: You ran mixed models with a lot of fixed variables but explain your results seemingly focusing just on single factors. This is unclear, also how you extracted the results, i.e. the final model after removing non-significant effects or the full model, and how you accounted for other significant effects. For example, you are stating the effects of reaction time without saying anything about the effects of light condition. It should be much clearer how you acquired the different results and the individual results should be more clearly and concisely spelled out. Also, one normally starts with explaining the highest interactions and then explaining lower level interactions and then individual fixed effects as post-hoc tests, which is not the way it is done here. Finally, you seem to have run a lot of statistical tests but did not correct for this, such as by using a Bonferroni correction.

Lines 194-197: This should be part of the methods, not the results.

Lines 199-200: This should already be stated in the introduction, not the results.

Lines 199-208: This paragraph is very hard to follow. Start by stating the results in such a way that it is clear what analysis you did, state the statistics, and then explain the result.

Lines 222-233: It is not clear if you presented the two types of cues in random order or if the olfactory cue always came second in the trial, which, if indeed the case, could just be an effect of timing.

Lines 227-228: Over what time period? Was this “after” period the same for both cue types?

Lines 228-231: hard to follow. Also a very specific result, taking two from the four population types in one light condition and looking at activity after the cue.

Line 237: As commented on before, it is still completely unclear what spring-fed light and glacial light entails and what it means to the fish and how that is relevant as compared to the difference in turbidity the fish may actually be experiencing.

Lines 242-247: Again a very specific result, and unclear. So you find a three-way interaction, but post-hoc tests actually showed only differences in activity existed for one habitat type? Also, what post-hoc test did you do here? This needs to be clear for each and every result you state.

Lines 250-251: What do you mean with ‘differences between cues’ and ‘their native environments’? The light treatments used?

Lines 256-269: Very complicated write-up again. Please make your results as clear as possible.

Lines 274-276: This detailed interpretation about evolutionary changes in use of sensory modalities and anti-predator behaviour does not follow naturally from the broader statement above that states just the existence of significant effects and interactions. This is more a statement for the concluding chapter. This initial paragraph would be a good place to provide a concise overview of the results, before going into detail discussing them.

Lines 281-283: Rewrite as currently “these differences” and “this reduced activity” do not refer to anything. Recapitulate concisely what you found before making your interpretations.

Lines 283-285: A couple things. First make clear how fish were different rather than just stating they were ‘more’ different. Second, why would this be ‘strong’ evidence? Third, “evolved differences for these highland fish”? You want to say that as fish populations that were more distant from the founding populations were more different this provides evidence for evolved

responses? Fourth, consider alternative explanations, such as could learning differences play a role or are there differences in how well the fish acclimatized to the lab/are used to disturbances.

Lines 290-291: As you didn't test this, I would stick to discussing your (many) results.

Lines 343-347: Is there any data on the genetic divergence on these or similar Icelandic stickleback populations or data that could inform the time it took for populations to establish the highland lakes?

Lines 366-370: How do you account for any learned or plastic behavioural responses of the fish?

Lines 373-374: This sentence is not clear.

Line 378: How do your behavioural results, linked to predator avoidance behaviour as you state yourself, help say something about colonization success? This is not directly clear.

Review form: Reviewer 2

Recommendation

Major revision is needed (please make suggestions in comments)

Scientific importance: Is the manuscript an original and important contribution to its field?

Good

General interest: Is the paper of sufficient general interest?

Good

Quality of the paper: Is the overall quality of the paper suitable?

Acceptable

Is the length of the paper justified?

Yes

Should the paper be seen by a specialist statistical reviewer?

No

Do you have any concerns about statistical analyses in this paper? If so, please specify them explicitly in your report.

No

It is a condition of publication that authors make their supporting data, code and materials available - either as supplementary material or hosted in an external repository. Please rate, if applicable, the supporting data on the following criteria.

Is it accessible?

Yes

Is it clear?

Yes

Is it adequate?

Yes

Do you have any ethical concerns with this paper?

No

Comments to the Author

Review of manuscript ID: RSPB-2021-0022.

In this manuscript the authors investigate how habitats varying in visibility, salinity and elevation affects anti-predator behaviour in three-spine sticklebacks. They test olfactory and mechano-visual predator cues on fish from four different habitats in two different light regimes over two years. They conclude that habitat did affect anti-predator behaviour and propose that their results suggest adaptive evolution and differences in behavioural plasticity in response to changes in visual environment.

In total, the authors tested almost 400 fish from 15 populations, and the manuscript indeed contains a lot of data (and hard work). The results are interesting both from a behavioural and evolutionary perspective, but also in a climate change context.

I have a couple of questions and concerns. Firstly, given that the manuscript contains a lot of data, it is a bit difficult for the reader to keep track of everything. In order to fully understand the methods and results, the reader has to read the supplementary material, which shouldn't be needed. This is most likely a product of the journal style of only allowing a shorter methods section in the main paper, but the experiment should still be understandable from the main text. For example, the collection sites (Fig S1), sample size and population characteristics (Table S1) and full results from the statistical models (Table S2-S6) are all found in the supplementary material.

Along this line, the authors state on line 143 that they used a sample size range of 20-41 per population. But that was then split in two light treatments, and some fish did not meet the criteria for inclusion during the experiments and had to be excluded, leaving a sample size as low as 8 for some populations. The actual sample size should be included in the main paper, and potential concerns regarding the robustness of the results due to low sample size should preferably be mentioned in the discussion.

The experiments were conducted in the summer, i.e. during the mating season. Although "non-gravid" adults were used (line 133), they were kept in the lab during 2-4 weeks prior to the experiments, which is ample time for sticklebacks to mature. Were the fish held in same-sex tanks during this time? If so, how did you ensure that they did not mate? When many males are held together during the mating season, they form dominance hierarchies with subordinate and dominant males exhibiting very different behavioural patterns. The sex of the test animals was confirmed by examining gonads, but sex is not included in the models, why? Given the likelihood that males and females differ in their behaviour, sex should at the very least be included in the models.

The authors also state that they checked presence of parasites (line 172), but nothing regarding parasite load is mentioned in the rest of the manuscript. Was there any difference in parasite load? If so that too should be included in the models. As with sex, parasites too can affect the behaviour of the fish, and sticklebacks can be very heavily parasitized. The authors themselves mention parasites as one of the factors that the fish needs to adapt to when colonizing new areas (line 63-66).

On line 171, it says that brains were dissected for subsequent gene expression analyses, but this is not mentioned any further, no such analyses or results are included in the manuscript. Why is this mentioned when not included?

For the chemical alarm cue experiment, it seems that there was no control? I.e., the fish would usually react to the water movement itself, and in order to control for this, a control treatment adding/injecting water with no cues are typically used. This cannot be corrected at this point, but the authors should acknowledge the fact that their experimental design does not allow to disentangle the fish reaction to water movement only from the presence of CAC (or CAC+

predator cues, as was used in 2018), and what this might potentially mean for the predator avoidance aspect of their experiment.

During the experiments, I assume altitude was the same for all fish? That would be the case if the university lab location corresponded to lowland. This means that while most habitat parameters were kept the same as where the population came from, all populations except for highland glacial were tested on the same altitude as where they came from. I.e., altitude was inevitably manipulated for the highland glacial population only? How do you think this might have affected the results?

There were many differences in experimental design between years. I understand that this was done because a lot of fish had to be excluded in 2017, and to make the fish more active in 2018 the authors added food cues during the experiment (it says in the supplementary material that: "In 2018, a transparent plastic bag containing three freshly thawed blood worms was attached to the end of the hose in the experimental tank to encourage fish to approach the location where the olfactory cue would be released"). They also added predator cues to the CAC cues, released the test fish directly into the experimental tank instead of using the chamber, used a different acclimation time etc. All of this information is found in the supplementary material but should be included in the main paper. Given the many differences in design between years, there is little surprise that year comes out as a strong factor in all models where it was included (Table S2-S5). This result is however not mentioned in the results section or mentioned or discussed in the discussion section. The figures also display the data pooled over years, but this is incorrect. The authors need to present and discuss the year effect, and present their data split by year, as per their results.

Decision letter (RSPB-2021-0022.R0)

01-Mar-2021

Dear Mrs Ålund:

I am writing to inform you that your manuscript RSPB-2021-0022 entitled "Sensory environment affects Icelandic threespine stickleback's anti-predator escape behaviour" has, in its current form, been rejected for publication in Proceedings B.

This action has been taken on the advice of referees, who have recommended that substantial revisions are necessary. With this in mind we would be happy to consider a resubmission, provided the comments of the referees are fully addressed. However please note that this is not a provisional acceptance.

- 1) A 'response to referees' document including details of how you have responded to the comments, and the adjustments you have made.

- 2) A clean copy of the manuscript and one with 'tracked changes' indicating your 'response to referees' comments document.
- 3) Line numbers in your main document.
- 4) Data - please see our policies on data sharing to ensure that you are complying (<https://royalsociety.org/journals/authors/author-guidelines/#data>).

Sincerely,
Dr Locke Rowe
mailto: proceedingsb@royalsociety.org

Associate Editor
Board Member: 1
Comments to Author:
Dear Authors,

I regret that I am unable to accept your paper for publication at Proc B. Both referees and myself have carefully read your manuscript and agree that you have collected an impressive data set that has high potential. However, I find myself in agreement with both reviewers who point out that results presented here are difficult to evaluate given lack of clarity in the methods and potential confounding factors. In addition, as referee 1 points out, the study's framing around anthropogenic changes, albeit highly attractive, is misleading as such conclusions do not arise from the natural populations evaluated here. The manuscript would greatly benefit from streamlining it. It is long and has information that is unclear how it relates to the study (e.g. brains collected for gene expression analysis).

Overall, critical issues such as methodological shortcomings (e.g. absence of controls for chemical cue experiments, lack of characterization of visual environment of testing conditions), missing information (e.g. final sample size per treatment per pop) and lack of clarity (eg. models in statistical analysis) seem to compromise the quality of this work.

I am sorry not to be able to give you a more positive answer, but I hope our referees' detailed comments prove useful should you decide to resubmit your manuscript here or send it elsewhere.

Best wishes
Ximena Bernal
Editor

Reviewer(s)' Comments to Author:
Referee: 1
Comments to the Author(s)

This study sought to investigate behavioural adaptations of Icelandic sticklebacks populations to the sensory environment of with different habitats. The authors collected a large number of sticklebacks from the wild from 15 different populations and tested them individually in the lab on their anti-predator behaviour. Fish were tested in two light conditions meant to resemble the clear and turbid waters of marine water/spring-fed lakes and glacial lakes respectively and both with a moving model predator and an olfactory cue. The authors found that habitat type, the light treatment, and predator cue all influenced the fish's anti-predator behaviour, often in complex ways.

I'd like to start by stating the authors did an impressive amount of work, testing close to 400 individual fish from many different populations and habitats. I also think there is potentially great value in the dataset the authors collected to help better understand behavioural adaptations. I have however a number of issues with the paper that warrant me from recommending it for publication.

First, the authors frame their paper in the context of human-induced environmental change but actually did not study behavioural responses to such change but to different natural habitats. It is fine to have a couple sentences in the discussion to state the potential relevance for such work, but framing the study in this way is not appropriate. It is also not needed as it is interesting enough to understand how the populations from the different natural habitats diverged from one another

Second, one of the main focus points of the paper seems to be the difference in turbidity between the habitat types. However, rather than testing fish in different levels of turbidity they tested fish in two different light conditions. The authors do not provide any clear reasoning for doing so nor enough detail to understand how fish perceive the light conditions or tests if the 'glacial light' condition indeed resembles turbidity by providing less visibility. It is also not considered that besides a reduced visibility, the change in brightness and colour of the light may give the fish a different sense of safety and the behavioural responses observed may in that case be more to do with the risk-taking behaviour than with sensory input.

Third, the experimental design was overly complex, with fish from four different habitat types tested with two light types and two odour cues. The authors thereby ran models with complex interactions and did a poor job in explaining the results, making it very hard to extract the key findings.

Fourth, the paper was really lengthy and far from to the point with for example the introduction being close to double the length of many ProcB papers. Also in general I found the authors did a poor job in explaining what they did and why, and in good order, such as stating experimental conditions before explaining the experimental design and very lengthy, hard to follow predictions without proper understanding of the experimental details. In combination with the complex interactions between the different factors and hard to follow results, the paper unfortunately does a poor job in explaining a potentially promising dataset.

Below I provide more detailed comments on the manuscript.

Lines 21-24: See my main point regarding the framing of the study in the context of environmental change.

Line 24: Why novel sensory environments? Fish moving to different habitats will experience different environmental conditions yes, but the environments are not necessarily novel (sensory speaking).

Line 28: Strictly speaking the population's habitat did not affect anti-predator behaviour but was linked to it. Or in other words, you found that fish's anti-predator behaviour could be explained by their habitat.

Lines 70-73: A more complex genetic architecture does not necessarily mean evolution is slower; it could also be faster.

Lines 77-78: This is not the place yet to state what was studied and should be moved down or better, explained in a different way.

Lines 79-81: This is important information that should come way earlier in the introduction. Actually, it would make more sense to me to write the paper around this focus rather than starting with the climate change narrative, which is not so fitting to the actual study.

Lines 86: What is meant with "current and historical" exactly? Make clear in the text.

Lines 86-87: If predation/predator evasion is suggested as a main potential selective pressure for sticklebacks in Iceland you should provide more detail about that here, such as the sympatry with predators and occurrence of which species. This is critical information currently missing from the manuscript (apart from a sentence in the discussion).

Lines 92-93: Again, information is provided about some part of the practical work before any real explanation of the study. Leave this for later in the introduction.

Line 100: burst swimming is not the same as a fast start and should not be confused with one another.

Lines 103-122: This is a paragraph of 288 words meant to explain the study and the predictions. This paragraph is really way too long and is hard to follow. Furthermore, despite its length, the study itself is actually still not well described (only 2 lines) and much remains unclear. For example, did you conduct an experimental or observational study? Was it done in the lab? What is meant with “light environments” and is this experimental design appropriate for investigating responses to turbidity? Etc. State concisely what you did exactly, with what purpose, and what predictions. The predictions do not need the long explanations given here when the proper background information is given above.

Lines 125-128: Here, the last three sentences of the introduction, the link is made again between turbidity and the potential relevance for human induced environmental change. This point is fine but is more for the end of the discussion, and framing the whole study in this context is not appropriate, see main point.

Line 133: You have to be more specific about where you caught the sticklebacks than “in Iceland”. Especially as you are testing fish from different types of sites and multiple replicates per site. I later found some of this information is provided in the SM. Integrate this and actually refer to that information.

Line 143: Clarify that tests were done with individual fish. How were these focal fish selected? How was the order of testing decided? Was population sufficiently randomized across trials? Much detail is missing about the experimental design.

Lines 147 onwards: Not sure why some words are underlined, but not a styling that ProcB supports as far as I know.

Lines 148-151: With what speed? Was this controlled and consistent across all trial? Was the replica already visible at the start of the trial or how was this prevented?

Lines 157-161: Given that turbidity is one of your main interests and main focus points of the study it is unclear to my why you not use turbidity itself in your experimental design but rather use a difference in “light environment” instead. Furthermore, you need to explain a lot more about these treatments, such as why you used light environments, what do the light environments entail, how are they perceived by the fish, and if pilots have been done to test if they are a good proxy for testing of turbidity. Did you run any tests to measure underwater visibility and was that lower for the ‘glacial light’ environment? Has such an experimental paradigm used before at all?

Lines 168-170: This is unclear. So each trial a fish received a second cue? Why was that done? Very likely the first cue will influence the second cue, making it much harder to disentangle the effects than were they tested separately. Also, was made sure the order of the cues was randomized?

Lines 175: What happened with the other 98 fish (~25% of the experimental fish)?

Line 185: So each fish received two trials? This is not clearly stated anywhere before, neither how much time there was between trials, if that was the same for all fish, etc.

Lines 186-187: You state interactions were included. But as you tested five fixed factors, what interactions did you actually include? And did you actually have some clear hypotheses for including all interactions?

Results: You ran mixed models with a lot of fixed variables but explain your results seemingly focusing just on single factors. This is unclear, also how you extracted the results, i.e. the final model after removing non-significant effects or the full model, and how you accounted for other significant effects. For example, you are stating the effects of reaction time without saying anything about the effects of light condition. It should be much clearer how you acquired the different results and the individual results should be more clearly and concisely spelled out. Also, one normally starts with explaining the highest interactions and then explaining lower level interactions and then individual fixed effects as post-hoc tests, which is not the way it is done here. Finally, you seem to have run a lot of statistical tests but did not correct for this, such as by using a Bonferroni correction.

Lines 194-197: This should be part of the methods, not the results.

Lines 199-200: This should already be stated in the introduction, not the results.

Lines 199-208: This paragraph is very hard to follow. Start by stating the results in such a way that it is clear what analysis you did, state the statistics, and then explain the result.

Lines 222-233: It is not clear if you presented the two types of cues in random order or if the olfactory cue always came second in the trial, which, if indeed the case, could just be an effect of timing.

Lines 227-228: Over what time period? Was this “after” period the same for both cue types?

Lines 228-231: hard to follow. Also a very specific result, taking two from the four population types in one light condition and looking at activity after the cue.

Line 237: As commented on before, it is still completely unclear what spring-fed light and glacial light entails and what it means to the fish and how that is relevant as compared to the difference in turbidity the fish may actually be experiencing.

Lines 242-247: Again a very specific result, and unclear. So you find a three-way interaction, but post-hoc tests actually showed only differences in activity existed for one habitat type? Also, what post-hoc test did you do here? This needs to be clear for each and every result you state.

Lines 250-251: What do you mean with ‘differences between cues’ and ‘their native environments’? The light treatments used?

Lines 256-269: Very complicated write-up again. Please make your results as clear as possible.

Lines 274-276: This detailed interpretation about evolutionary changes in use of sensory modalities and anti-predator behaviour does not follow naturally from the broader statement above that states just the existence of significant effects and interactions. This is more a statement for the concluding chapter. This initial paragraph would be a good place to provide a concise overview of the results, before going into detail discussing them.

Lines 281-283: Rewrite as currently “these differences” and “this reduced activity” do not refer to anything. Recapitulate concisely what you found before making your interpretations.

Lines 283-285: A couple things. First make clear how fish were different rather than just stating they were ‘more’ different. Second, why would this be ‘strong’ evidence? Third, “evolved differences for these highland fish”? You want to say that as fish populations that were more distant from the founding populations were more different this provides evidence for evolved responses? Fourth, consider alternative explanations, such as could learning differences play a role or are there differences in how well the fish acclimatized to the lab/are used to disturbances.

Lines 290-291: As you didn’t test this, I would stick to discussing your (many) results.

Lines 343-347: Is there any data on the genetic divergence on these or similar Icelandic stickleback populations or data that could inform the time it took for populations to establish the highland lakes?

Lines 366-370: How do you account for any learned or plastic behavioural responses of the fish?

Lines 373-374: This sentence is not clear.

Line 378: How do your behavioural results, linked to predator avoidance behaviour as you state yourself, help say something about colonization success? This is not directly clear.

Referee: 2

Comments to the Author(s)

Review of manuscript ID: RSPB-2021-0022.

In this manuscript the authors investigate how habitats varying in visibility, salinity and elevation affects anti-predator behaviour in three-spine sticklebacks. They test olfactory and mechano-visual predator cues on fish from four different habitats in two different light regimes over two years. They conclude that habitat did affect anti-predator behaviour and propose that their results suggest adaptive evolution and differences in behavioural plasticity in response to changes in visual environment.

In total, the authors tested almost 400 fish from 15 populations, and the manuscript indeed contains a lot of data (and hard work). The results are interesting both from a behavioural and evolutionary perspective, but also in a climate change context.

I have a couple of questions and concerns. Firstly, given that the manuscript contains a lot of data, it is a bit difficult for the reader to keep track of everything. In order to fully understand the methods and results, the reader has to read the supplementary material, which shouldn’t be

needed. This is most likely a product of the journal style of only allowing a shorter methods section in the main paper, but the experiment should still be understandable from the main text. For example, the collection sites (Fig S1), sample size and population characteristics (Table S1) and full results from the statistical models (Table S2-S6) are all found in the supplementary material.

Along this line, the authors state on line 143 that they used a sample size range of 20-41 per population. But that was then split in two light treatments, and some fish did not meet the criteria for inclusion during the experiments and had to be excluded, leaving a sample size as low as 8 for some populations. The actual sample size should be included in the main paper, and potential concerns regarding the robustness of the results due to low sample size should preferably be mentioned in the discussion.

The experiments were conducted in the summer, i.e. during the mating season. Although “non-gravid” adults were used (line 133), they were kept in the lab during 2-4 weeks prior to the experiments, which is ample time for sticklebacks to mature. Were the fish held in same-sex tanks during this time? If so, how did you ensure that they did not mate? When many males are held together during the mating season, they form dominance hierarchies with subordinate and dominant males exhibiting very different behavioural patterns. The sex of the test animals was confirmed by examining gonads, but sex is not included in the models, why? Given the likelihood that males and females differ in their behaviour, sex should at the very least be included in the models.

The authors also state that they checked presence of parasites (line 172), but nothing regarding parasite load is mentioned in the rest of the manuscript. Was there any difference in parasite load? If so that too should be included in the models. As with sex, parasites too can affect the behaviour of the fish, and sticklebacks can be very heavily parasitized. The authors themselves mention parasites as one of the factors that the fish needs to adapt to when colonizing new areas (line 63-66).

On line 171, it says that brains were dissected for subsequent gene expression analyses, but this is not mentioned any further, no such analyses or results are included in the manuscript. Why is this mentioned when not included?

For the chemical alarm cue experiment, it seems that there was no control? I.e., the fish would usually react to the water movement itself, and in order to control for this, a control treatment adding/injecting water with no cues are typically used. This cannot be corrected at this point, but the authors should acknowledge the fact that their experimental design does not allow to disentangle the fish reaction to water movement only from the presence of CAC (or CAC+ predator cues, as was used in 2018), and what this might potentially mean for the predator avoidance aspect of their experiment.

During the experiments, I assume altitude was the same for all fish? That would be the case if the university lab location corresponded to lowland. This means that while most habitat parameters were kept the same as where the population came from, all populations except for highland glacial were tested on the same altitude as where they came from. I.e., altitude was inevitably manipulated for the highland glacial population only? How do you think this might have affected the results?

There were many differences in experimental design between years. I understand that this was done because a lot of fish had to be excluded in 2017, and to make the fish more active in 2018 the authors added food cues during the experiment (it says in the supplementary material that: “In 2018, a transparent plastic bag containing three freshly thawed blood worms was attached to the end of the hose in the experimental tank to encourage fish to approach the location where the olfactory cue would be released”). They also added predator cues to the CAC cues, released the test fish directly into the experimental tank instead of using the chamber, used a different

acclimation time etc. All of this information is found in the supplementary material but should be included in the main paper. Given the many differences in design between years, there is little surprise that year comes out as a strong factor in all models where it was included (Table S2-S5). This result is however not mentioned in the results section or mentioned or discussed in the discussion section. The figures also display the data pooled over years, but this is incorrect. The authors need to present and discuss the year effect, and present their data split by year, as per their results.

Author's Response to Decision Letter for (RSPB-2021-0022.R0)

See Appendix A.

RSPB-2021-1881.R0

Review form: Reviewer 1

Recommendation

Major revision is needed (please make suggestions in comments)

Scientific importance: Is the manuscript an original and important contribution to its field?

Good

General interest: Is the paper of sufficient general interest?

Good

Quality of the paper: Is the overall quality of the paper suitable?

Good

Is the length of the paper justified?

Yes

Should the paper be seen by a specialist statistical reviewer?

No

Do you have any concerns about statistical analyses in this paper? If so, please specify them explicitly in your report.

No

It is a condition of publication that authors make their supporting data, code and materials available - either as supplementary material or hosted in an external repository. Please rate, if applicable, the supporting data on the following criteria.

Is it accessible?

Yes

Is it clear?

Yes

Is it adequate?

Yes

Do you have any ethical concerns with this paper?

No

Comments to the Author

I previously reviewed this manuscript and think the authors responded well to my and the other reviewer's concerns and much improved their manuscript based on our feedback. The considerably revised text is more focused and reads well. The authors also better now provide more background information and explanation regarding the light treatments and better re-framed the study. I still however have two main concerns as well as some other comments that should be addressed.

My first main concern is that what you are doing with your light treatment is simply comparing fish's anti-predator behaviour at normal light versus slightly dimmed light. Although higher turbidity goes together with faster declines in brightness, only using a difference in light intensity (and color) is not the same as a difference in nr of suspended particles. The aquatic environment may be darker but will have a similar level of transparency. Furthermore, looking at Figure S1 the difference in brightness between the treatments actually seems quite minor. I am therefore still questioning a bit how relevant your study design is to look at anti-predator behaviour in the context of the differences between the habitats in the wild. Also, the secchi distances you document for the glacial lakes were 0.8, 1.26 and 1.6m, which is considerably more than the response distances the fish showed to the fake predator.

My second main concern is that your effect sizes seem rather weak. In your manuscript you mostly focus on the existence of significant effects, but the stated differences in the results and those visible in the figures suggest the effects are very limited, with huge overlap between the fish from the different habitat. This questions a bit how important the effects are that you are describing and at least you should better state the effect sizes in your results and discuss their relevance in your discussion.

Other comments:

- Abstract: You should revise your abstract to more clearly spell out the key results. Currently, you talk about "reactions differed" and "fish from different habitats differed in their reactions" and then immediately go to the interpretations, which therefore do not naturally follow. You also end with four sentences in which you are basically restating what you said in the sentences before, valuable space you could use there to state the specific results. This is especially needed because of the complex experimental design used.

- You have data of two years but didn't clarify how your 15 waterbodies were allocated across the two years. Did you for example get fish from all 15 waterbodies in each year or 7 in one year and 8 from the other? If more the latter, was made sure that habitats were equally sampled across both years?

- Lines 111-112: You state to expect the biggest differences between spring fed and glacial habitats. As those are the only two habitats you tested it is unclear what you are trying to say here.

- Lines 131-137: you calculate percent visibility by dividing secchi depth by local lake depth. This is wrong as it would result in a deeper lake having a lower visibility. Looking at the supplementary I understand you tried this approach to account for many of the lakes being too shallow to lose sight of the secchi disk. Therefore an alternative approach would have been preferred to measure water transparency. Since that is not possible anymore without going out again and remeasuring, I suggest to just use a categorical variable.

- Line 169: "Live scoring". How many unique observers contributed to this and were they naive in terms of knowing the habitat the fish came from?

- Line 228: a slower response not necessarily means they are slower at detecting. Rephrase.
- Figure 1. For B and C, why do you only show data for two of the lakes? In C it remains unclear what the different lines are, i.e. what depths they correspond to. From C it appears that the Glacial light treatment does not correspond well to the irradiance measured in the glacier lake. With D, I appreciate your attempt to clarify your study design but this figure doesn't really add much and would not have that in the main text. Also note that your font size is far from consistent across the panels a-d and that the labels are far too large.
- Figure 2 is hard to comprehend. Readers should within seconds be able to see the main result from the figure. Now the reader has to first read the axes, then the labels, then go over the pie charts, then look at the legend, to then go back to the pie charts to start and see how they are different. Pie charts are not ideal for comparisons nor show differences in the spread of the data.

Review form: Reviewer 2

Recommendation

Accept with minor revision (please list in comments)

Scientific importance: Is the manuscript an original and important contribution to its field?

Good

General interest: Is the paper of sufficient general interest?

Good

Quality of the paper: Is the overall quality of the paper suitable?

Good

Is the length of the paper justified?

Yes

Should the paper be seen by a specialist statistical reviewer?

No

Do you have any concerns about statistical analyses in this paper? If so, please specify them explicitly in your report.

Yes

It is a condition of publication that authors make their supporting data, code and materials available - either as supplementary material or hosted in an external repository. Please rate, if applicable, the supporting data on the following criteria.

Is it accessible?

Yes

Is it clear?

Yes

Is it adequate?

Yes

Do you have any ethical concerns with this paper?

No

Comments to the Author

Review of the revised version of manuscript RSPB-2021-1881. The authors have made a great effort to revise the manuscript and it is now much clearer regarding many of my previous comments. In particular the inclusion of new Figure 1 makes the methods and sample size readily available to the reader. For some of my previous comments, the authors provided a response but did not add or change anything in the manuscript, although in some instances the information provided in the response needs to be included in the manuscript. See my specific comments regarding this below. In particular, I find it odd that the authors almost completely ignore a significant effect because they do not concern it to be of interest to the study (line 540). In their response the authors say that the between-year differences do not affect the results presented regarding habitat types, predator cues and light treatments. But how can the authors know this? The potential interactions between year and any other fixed effect are never tested (see my last comment).

Note that the line numbers below refer to line numbers in the word manuscript file with track changes, and not the proof pdf.

Line 231: Predation can be a strong selective pressure for wild fish irrespective of their size, as long as large enough predators are around! I'd hence remove the word "small" from this sentence.

Line 233-235: It would be appropriate to cite some of Ulrika Candolins work here, since they have shown that stickleback do rely on other cues (olfactory) than visual under turbid conditions. This was mainly tested in a sexual selection context, but is valid here and supports your expectation. Candolin 2009 provides an overview, specifically see the section in that paper with the subheading "switch to other cues"

Candolin U. 2009. Population responses to anthropogenic disturbance: lessons from three-spined sticklebacks *Gasterosteus aculeatus* in eutrophic habitats. *J. Fish Biol.*

Line 288-290: I previously asked the authors regarding housing of fish prior to the experiments, and the effect of sex, given that the experiments were performed during summer, i.e., the stickleback's mating season. The authors did provide a response but as far as I can tell, some of the information was not added to the manuscript. The authors need to add to the manuscript that the fish were kept in mixed-sex tanks and that gravid females were avoided when selecting fish for the experiments (as per their response to my previous comment). The authors did not respond to my comment regarding dominance hierarchies in males, something that occur when keeping stickleback males together during the mating season. Since it has been added to the revised version of the manuscript that the authors did not make any specific selection regarding what fish to use in the experiment (line 298 states that fish were selected haphazardly, which isn't completely true for females then, since the authors responded that gravid females were avoided), I assume that the use of dominant and subordinate males in the different treatment groups would have been at random.

I suggest adding the information regarding mixed-sex tanks to line 290, for example like this: "and housed 1-4 weeks in 13 litre opaque tanks by population (i.e., in mixed-sex tanks)"

I suggest that the authors add to line 298 that gravid females were avoided:

"Each fish, selected haphazardly from its holding tank (note however that gravid females were avoided)"

Line 289: Add that the university laboratory was at a lowland location. The authors responded to my previous comment regarding the fact that all populations except for highland glacial were tested on the same altitude as where they came from (i.e., meaning that altitude was inevitably manipulated for the highland glacial population only), but they did not add anything to the

manuscript. At the very least, the information that the laboratory was at a lowland location should be available to the readers of this paper.

Similarly to the above, the authors responded to my comment regarding lack of control (i.e., only water) in the CAC and CAC+predator cue treatments, but they did not add anything to the manuscript. I think their line of reasoning is OK, but it should be provided to the readers and not only to reviewers. Given the space limitations in the main paper I would add this to the supplementary material, where their pilot trials and the other aspects of not including a control can be mentioned.

Regarding my comment on the wear effect, the authors have now clarified this in the statistical analyses section of the main paper that year had an effect, and further write that “but are not of specific interest for hypotheses tested in this study” (line 538-540). While it is good that the authors at least mention the year effect in the main paper (although it would have been better suited to do so in the result section rather than in the statistical analyses section), it is incorrect to almost completely ignore an effect because the authors do not concern it to be of interest to the study. In their response the authors say that “year” is accounted for because it is included in the model, and that the between-year differences do not take away the results presented regarding habitat types, predator cues and light treatments. But how can the authors know this? As far as I can tell from the models (Tables S4, S5, and all of table S7), the interaction between year and any other fixed effect is never tested.

Decision letter (RSPB-2021-1881.R0)

06-Oct-2021

I am writing to inform you that this version of your manuscript RSPB-2021-1881 entitled "Sensory environment affects Icelandic threespine stickleback's anti-predator escape behaviour" has, in its current form, been rejected for publication in Proceedings B.

This action has been taken on the advice of referees, who have recommended that substantial revisions are necessary. With this in mind we would be happy to consider a resubmission, provided the comments of the referees are fully addressed. However please note that this is not a provisional acceptance.

Please find below the comments made by the referees, not including confidential reports to the Editor, which I hope you will find useful.

4) Please read our data sharing policies to ensure that you meet our requirements <https://royalsociety.org/journals/authors/author-guidelines/#data>.

Sincerely,
Dr Locke Rowe
mailto: proceedingsb@royalsociety.org

Associate Editor Board Member

Comments to Author:

The authors of this manuscript have done a good job at addressing the inquiries raised. The explanations are clear and help clarify previously confusing points. Several clarifications though should be included in the manuscript as concerns similar to those raised by the reviewers will likely come up among the readers.

The experimental design is now easier to follow and an extension of that is that additional queries and concerns have been raised about the study. In particular, one of the reviewers validly questions whether the experiment to address the effect of turbidity is indeed getting at that. The relevance of this study design needs to be stronger and better connected to the point at hand (turbidity) for a solid interpretation of the findings. An additional but also highly relevant point reflects concerns about the small effect size of the findings. While the results may be significant, the effect sizes should be clearly stated and fully discussed in the study. Finally, reviewers provide excellent suggestions to improve the manuscript including rewording of the abstract and redesigning Figure 2 that is currently difficult to digest.

Reviewer(s)' Comments to Author:

Referee: 2

Comments to the Author(s).

Review of the revised version of manuscript RSPB-2021-1881. The authors have made a great effort to revise the manuscript and it is now much clearer regarding many of my previous comments. In particular the inclusion of new Figure 1 makes the methods and sample size readily available to the reader. For some of my previous comments, the authors provided a response but did not add or change anything in the manuscript, although in some instances the information provided in the response needs to be included in the manuscript. See my specific comments regarding this below. In particular, I find it odd that the authors almost completely ignore a significant effect because they do not concern it to be of interest to the study (line 540). In their response the authors say that the between-year differences do not affect the results presented regarding habitat types, predator cues and light treatments. But how can the authors know this? The potential interactions between year and any other fixed effect are never tested (see my last comment).

Note that the line numbers below refer to line numbers in the word manuscript file with track changes, and not the proof pdf.

Line 231: Predation can be a strong selective pressure for wild fish irrespective of their size, as long as large enough predators are around! I'd hence remove the word "small" from this sentence.

Line 233-235: It would be appropriate to cite some of Ulrika Candolins work here, since they have shown that stickleback do rely on other cues (olfactory) than visual under turbid conditions. This was mainly tested in a sexual selection context, but is valid here and supports your expectation.

Candolin 2009 provides an overview, specifically see the section in that paper with the subheading “switch to other cues”

Candolin U. 2009. Population responses to anthropogenic disturbance: lessons from three-spined sticklebacks *Gasterosteus aculeatus* in eutrophic habitats. *J. Fish Biol.*

Line 288-290: I previously asked the authors regarding housing of fish prior to the experiments, and the effect of sex, given that the experiments were performed during summer, i.e., the stickleback’s mating season. The authors did provide a response but as far as I can tell, some of the information was not added to the manuscript. The authors need to add to the manuscript that the fish were kept in mixed-sex tanks and that gravid females were avoided when selecting fish for the experiments (as per their response to my previous comment). The authors did not respond to my comment regarding dominance hierarchies in males, something that occur when keeping stickleback males together during the mating season. Since it has been added to the revised version of the manuscript that the authors did not make any specific selection regarding what fish to use in the experiment (line 298 states that fish were selected haphazardly, which isn’t completely true for females then, since the authors responded that gravid females were avoided), I assume that the use of dominant and subordinate males in the different treatment groups would have been at random.

I suggest adding the information regarding mixed-sex tanks to line 290, for example like this:

“and housed 1-4 weeks in 13 litre opaque tanks by population (i.e., in mixed-sex tanks)”

I suggest that the authors add to line 298 that gravid females were avoided:

“Each fish, selected haphazardly from its holding tank (note however that gravid females were avoided)”

Line 289: Add that the university laboratory was at a lowland location. The authors responded to my previous comment regarding the fact that all populations except for highland glacial were tested on the same altitude as where they came from (i.e., meaning that altitude was inevitably manipulated for the highland glacial population only), but they did not add anything to the manuscript. At the very least, the information that the laboratory was at a lowland location should be available to the readers of this paper.

Similarly to the above, the authors responded to my comment regarding lack of control (i.e., only water) in the CAC and CAC+predator cue treatments, but they did not add anything to the manuscript. I think their line of reasoning is OK, but it should be provided to the readers and not only to reviewers. Given the space limitations in the main paper I would add this to the supplementary material, where their pilot trials and the other aspects of not including a control can be mentioned.

Regarding my comment on the wear effect, the authors have now clarified this in the statistical analyses section of the main paper that year had an effect, and further write that “but are not of specific interest for hypotheses tested in this study” (line 538-540). While it is good that the authors at least mention the year effect in the main paper (although it would have been better suited to do so in the result section rather than in the statistical analyses section), it is incorrect to almost completely ignore an effect because the authors do not concern it to be of interest to the study. In their response the authors say that “year” is accounted for because it is included in the model, and that the between-year differences do not take away the results presented regarding habitat types, predator cues and light treatments. But how can the authors know this? As far as I can tell from the models (Tables S4, S5, and all of table S7), the interaction between year and any other fixed effect is never tested.

Referee: 1

Comments to the Author(s).

I previously reviewed this manuscript and think the authors responded well to my and the other reviewer's concerns and much improved their manuscript based on our feedback. The considerably revised text is more focused and reads well. The authors also better now provide more background information and explanation regarding the light treatments and better re-

framed the study. I still however have two main concerns as well as some other comments that should be addressed.

My first main concern is that what you are doing with your light treatment is simply comparing fish's anti-predator behaviour at normal light versus slightly dimmed light. Although higher turbidity goes together with faster declines in brightness, only using a difference in light intensity (and color) is not the same as a difference in nr of suspended particles. The aquatic environment may be darker but will have a similar level of transparency. Furthermore, looking at Figure S1 the difference in brightness between the treatments actually seems quite minor. I am therefore still questioning a bit how relevant your study design is to look at anti-predator behaviour in the context of the differences between the habitats in the wild. Also, the secchi distances you document for the glacial lakes were 0.8, 1.26 and 1.6m, which is considerably more than the response distances the fish showed to the fake predator.

My second main concern is that your effect sizes seem rather weak. In your manuscript you mostly focus on the existence of significant effects, but the stated differences in the results and those visible in the figures suggest the effects are very limited, with huge overlap between the fish from the different habitat. This questions a bit how important the effects are that you are describing and at least you should better state the effect sizes in your results and discuss their relevance in your discussion.

Other comments:

- Abstract: You should revise your abstract to more clearly spell out the key results. Currently, you talk about "reactions differed" and "fish from different habitats differed in their reactions" and then immediately go to the interpretations, which therefore do not naturally follow. You also end with four sentences in which you are basically restating what you said in the sentences before, valuable space you could use there to state the specific results. This is especially needed because of the complex experimental design used.

- You have data of two years but didn't clarify how your 15 waterbodies were allocated across the two years. Did you for example get fish from all 15 waterbodies in each year or 7 in one year and 8 from the other? If more the latter, was made sure that habitats were equally sampled across both years?

- Lines 111-112: You state to expect the biggest differences between spring fed and glacial habitats. As those are the only two habitats you tested it is unclear what you are trying to say here.

- Lines 131-137: you calculate percent visibility by dividing secchi depth by local lake depth. This is wrong as it would result in a deeper lake having a lower visibility. Looking at the supplementary I understand you tried this approach to account for many of the lakes being too shallow to lose sight of the secchi disk. Therefore an alternative approach would have been preferred to measure water transparency. Since that is not possible anymore without going out again and remeasuring, I suggest to just use a categorical variable.

- Line 169: "Live scoring". How many unique observers contributed to this and were they naive in terms of knowing the habitat the fish came from?

- Line 228: a slower response not necessarily means they are slower at detecting. Rephrase.

- Figure 1. For B and C, why do you only show data for two of the lakes? In C it remains unclear what the different lines are, i.e. what depths they correspond to. From C it appears that the Glacial light treatment does not correspond well to the irradiance measured in the glacier lake. With D, I appreciate your attempt to clarify your study design but this figure doesn't really add much and would not have that in the main text. Also note that your font size is far from consistent across the panels a-d and that the labels are far too large.

- Figure 2 is hard to comprehend. Readers should within seconds be able to see the main result from the figure. Now the reader has to first read the axes, then the labels, then go over the pie charts, then look at the legend, to then go back to the pie charts to start and see how they are different. Pie charts are not ideal for comparisons nor show differences in the spread of the data.

Author's Response to Decision Letter for (RSPB-2021-1881.R0)

See Appendix B.

RSPB-2022-0044.R0

Review form: Reviewer 2

Recommendation

Accept as is

Scientific importance: Is the manuscript an original and important contribution to its field?

Good

General interest: Is the paper of sufficient general interest?

Good

Quality of the paper: Is the overall quality of the paper suitable?

Good

Is the length of the paper justified?

Yes

Should the paper be seen by a specialist statistical reviewer?

No

Do you have any concerns about statistical analyses in this paper? If so, please specify them explicitly in your report.

No

It is a condition of publication that authors make their supporting data, code and materials available - either as supplementary material or hosted in an external repository. Please rate, if applicable, the supporting data on the following criteria.

Is it accessible?

Yes

Is it clear?

Yes

Is it adequate?

Yes

Do you have any ethical concerns with this paper?

No

Comments to the Author

Review of the 2:nd revision of manuscript RSPB-2022-0044. The authors have replied to my outstanding questions and comments in a satisfactory manner. They have added the information I requested (that was provided in their responses to my previous comments but not added to the former version of the MS). I have no further comments that needs to be addressed. The authors have done a great job with this manuscript, and it provides important data to this field.

I happened to note that the manuscript has been given a new manuscript number each time it was submitted, I assume this is because it was rejected, although allowing the authors to submit a revised version. Despite this, I assume that Proceedings of the Royal Society B will use the original date submitted, i.e., January 5, 2021, as the official received date. Using the date for any of the resubmissions would be fiddling with the journals data on handling time.

Decision letter (RSPB-2022-0044.R0)

09-Feb-2022

Dear Mrs Ålund:

Your manuscript has now been peer reviewed and the reviews have been assessed by an Associate Editor. The reviewers' comments (not including confidential comments to the Editor) and the comments from the Associate Editor are included at the end of this email for your reference. As you will see, the reviewers and the Editors have raised some concerns with your manuscript and we would like to invite you to revise your manuscript to address them.

Research ethics:

Use of animals and field studies:

It is a condition of publication that you make available the data and research materials supporting the results in the article (<https://royalsociety.org/journals/authors/author-guidelines/#data>). Datasets should be deposited in an appropriate publicly available repository and details of the associated accession number, link or DOI to the datasets must be included in the Data Accessibility section of the article (<https://royalsociety.org/journals/ethics-policies/data-sharing-mining/>). Reference(s) to datasets should also be included in the reference list of the article with DOIs (where available).

Please submit a copy of your revised paper within three weeks. If we do not hear from you within this time your manuscript will be rejected. If you are unable to meet this deadline please let us know as soon as possible, as we may be able to grant a short extension.

Best wishes,
 Dr Locke Rowe
 mailto: proceedingsb@royalsociety.org

Associate Editor Board Member

Comments to Author:

The manuscript has improved tremendously in clarity and I comment the authors for their work revising this work. This is an interesting study that uses an impressive data set. However, the experimental design does not allow the authors to tease apart whether the changes they are seeing in anti-predatory responses are due to the effect of early experience and/or genetic changes/adaptations. As only adults were tested, early experience could explain the differences in antipredatory behavior found here. The conclusions about rapid adaptation in antipredatory strategies is thus premature in this system.

The manuscript takes big leaps in concluding that rapid adaptation is at place without solid evidence this is the case. While I see the charm associated with rapid adaptation, acknowledgment of the limitations of the study is necessary. Given that the study is framed around rapid environmental changes, the role of early experience is also interesting. Even with this limitation, the study still provides valuable insights that will be of interested of ecologists and evolutionary biologist.

Specific comments:

Abstract: conclusions of rapid adaptation need to be tone down.

Line 43: change to "habitats are rapidly changing because of"

Line 67: what to you mean by "these issues"? be more specific.

Line 71: add a citation to support the claim of adaptations in physiology

Line 76: provide a reference value for "higher elevation", >400masl

Line 104: rather than calling them "alternative", it would be more straightforward to call this the "novel visual environment"

Lines 111-113: it would be helpful for the reader to add the rationale behind the first prediction, like you did on prediction 3. While the reader may be able to guess the thought process that took you to such predictions, for the sake of clarity it would be better to flesh it out.

Line 114, prediction 2, do you mean the magnitude of the difference depends on their native sensory environment?

Line 161-162, change to: "...an equivalent reduction in light as the one experienced by as fish in a spring-fed lake..."

Line 208, what predictions were those? It is unclear what the authors mean by 'below' and how those predictions are different from the ones in the introduction.

Line 304, even though the sample size is impressive, and I understand the authors are proud of it, the number of fish and populations have already been highlighted before. I suggest de-removing it from here and focusing on the conceptual gain of the study.

Lines 310-311, incomplete sentence. Needs to be re-written, here is a suggestion I think encapsulates the idea of the authors: "our comparison addresses lakes at different elevations and thus involves populations separated over many generations suggesting some evolutionary change". However, this is not a convincing argument for evolutionary change. Even though it seems likely, the authors cannot rule out the role of early experience. Indeed, there could be a complex interaction between experience and genetic differences among populations shaping the findings shown here. The authors should more transparently discuss the speculative nature of the conclusion of evolutionary changes specific to antipredatory behavior.

Lines 331-333. Rapid evolution may be happening but further work that confirms this hypothesis is necessary as the current study does not scaffold the effect of experience vs genetic responses. This sentence needs to put in context. A common garden experimental design would be necessary to arise to the conclusions posed here.

Lines 373-375, these results are also in line with the idea that experience may be shaping such antipredatory behaviors as the highland fish seem to live in low predation environments.

Reviewer(s)' Comments to Author:

Referee: 2

Comments to the Author(s).

Review of the 2:nd revision of manuscript RSPB-2022-0044. The authors have replied to my outstanding questions and comments in a satisfactory manner. They have added the information I requested (that was provided in their responses to my previous comments but not added to the former version of the MS). I have no further comments that needs to be addressed. The authors have done a great job with this manuscript, and it provides important data to this field.

Author's Response to Decision Letter for (RSPB-2022-0044.R0)

See Appendix C.

Decision letter (RSPB-2022-0044.R1)

08-Mar-2022

Dear Dr Ålund

I am pleased to inform you that your manuscript entitled "Sensory environment affects Icelandic threespine stickleback's anti-predator escape behaviour" has been accepted for publication in Proceedings B.

Data Accessibility section

Open Access

Paper charges

Sincerely,

Dr Locke Rowe

Associate Editor:

Comments to Author:

Excellent job at revising this work! More clarity about the goals and implications of the high vs low elevations populations makes the text easier to follow. More importantly, the more transparent and open discussion both about the potential mechanisms driving the observed changes (evolution and/or plasticity) have eliminated any concerns I had about the manuscript and made the discussion much stronger. Overall, the manuscript has tremendously improved and I am looking forward to seeing it published.

Appendix A

Editor's comments:

Both referees and myself have carefully read your manuscript and agree that you have collected an impressive data set that has high potential. However, I find myself in agreement with both reviewers who point out that results presented here are difficult to evaluate given lack of clarity in the methods and potential confounding factors. In addition, as referee 1 points out, the study's framing around anthropogenic changes, albeit highly attractive, is misleading as such conclusions do not arise from the natural populations evaluated here. The manuscript would greatly benefit from streamlining it. It is long and has information that is unclear how it relates to the study (e.g. brains collected for gene expression analysis).

Response: Thank you very much for your interest, concrete feedback and the opportunity to improve this manuscript. The new manuscript version we submit here has been extensively rewritten. We have made major changes in how we introduce and discuss our study, including the description of the context and goals of the experiment. In addition, we have added more detail to clarify both the methods and results sections. As a result, both the introduction and discussion sections are now considerably shorter and more to the point, while the method section has been slightly expanded to move more details into the main text from the supplement. The manuscript text as a whole is thus shorter, although we have added a new figure (Figure 1) that shows the sampling locations, a quantitative description of the light treatments and how they compare to the natural habitats, and the experimental design of the study, including additional information about sample sizes. While including an additional figure will require space, this is offset by an overall reduction in number of words in the manuscript text. In addition, we believe this figure will highly improve the clarity of the paper and it addresses a number of the reviewer questions in a visual format.

We have now reframed the context of the study, focussing more on colonization and rapid adaptation to novel habitats *per se*, and on the strengths of our study system, which provides a rare opportunity to compare multiple populations for each of the studied environments, and thus allows us to get insights into the repeatability of behavioural changes in response to high turbidity and associated lower visibility. As we point out on current line 53, new glacial lakes are being formed at an increasing rate because of the accelerated melting and retreating of glaciers, indirectly caused by human-induced global warming. Hence this study is relevant in the context of current and future scenarios of anthropogenetic environmental disturbances. We have tried to clarify this point in the introduction. We agree, however, that we do not provide direct measures of human impacts for the specific individual lakes we studied, and have thus toned down this narrative throughout the manuscript.

Overall, critical issues such as methodological shortcomings (e.g. absence of controls for chemical cue experiments, lack of characterization of visual environment of testing

conditions), missing information (e.g. final sample size per treatment per pop) and lack of clarity (eg. models in statistical analysis) seem to compromise the quality of this work.

Response: The methods section has now been expanded to clarify these issues, that are also discussed in more detail below in our point-by-point response to the reviewers' comments. Several of the items in the list have now been moved from the supplemental material to our new Figure 1 so that they are more obvious, specifically the characterization of the light environment both in nature and in the lab experiment and the sample sizes for treatment groups. The visual environment in the experimental tanks was designed to best represent changes in brightness measured directly in representative glacial lakes using spectrophotometry (see Figure 1b-c and line 133). The final sample sizes for each population are given in the new Figure 1a on a map of the population locations. The mean and standard error for the final sample sizes of each population are broken down by each light treatment x cue type x cue order in the new Figure 1d with sample size information about each habitat type x light treatment x cue type given in the figure legend. Additional information about sample sizes can be found in the supplement, now in its own table, Table S2. Details about the statistical analyses have been edited in the methods (starting line 184).

We greatly welcomed the opportunity to revise this manuscript and we believe you will find we have used the comments provided by yourself and the two reviewers to substantially improve the manuscript. Thank you to you and the reviewers for helping us make this a better paper.

Referee: 1

This study sought to investigate behavioural adaptations of Icelandic sticklebacks populations to the sensory environment of with different habitats. The authors collected a large number of sticklebacks from the wild from 15 different populations and tested them individually in the lab on their anti-predator behaviour. Fish were tested in two light conditions meant to resemble the clear and turbid waters of marine water/spring-fed lakes and glacial lakes respectively and both with a moving model predator and an olfactory cue. The authors found that habitat type, the light treatment, and predator cue all influenced the fish's anti-predator behaviour, often in complex ways.

I'd like to start by stating the authors did an impressive amount of work, testing close to 400 individual fish from many different populations and habitats. I also think there is potentially great value in the dataset the authors collected to help better understand behavioural adaptations. I have however a number of issues with the paper that warrant me from recommending it for publication.

Response: Thank you for your assessment of our work and of the potential of our dataset and manuscript, and for the suggestions that we believe greatly helped in making this new version much clearer. See below for the detailed response to your individual comments.

First, the authors frame their paper in the context of human-induced environmental change but actually did not study behavioural responses to such change but to different natural habitats. It is fine to have a couple sentences in the discussion to state the potential relevance for such work, but framing the study in this way is not appropriate. It is also not needed as it is interesting enough to understand how the populations from the different natural habitats diverged from one another

Response: We agree that the study of the different habitats we focus on is interesting on its own and have now reframed the manuscript to directly highlight the strengths of the Icelandic network of marine and freshwater habitats for studying repeated adaptation to similar ecological conditions, starting in the introduction. We are now also clearer in our description of how this study relates to climate change more globally, for example specifying on current line 53 that glacial lakes are expected to be formed at an increasing pace given the current and predicted future rate at which glaciers are melting and retreating. Although the glacial lakes we studied are indeed natural, many were formed recently because of the retreat of glaciers due to climate change. Icelanders have detailed records of glacial retreat as well as can determine the time when some lakes were formed, and some are within the past 50 years. Stickleback colonizing them from the ocean encounter quite a novel sensory environment, and this is partly due to human disturbance. Encountering such novelty is part of the motivation for our study. We tried to clarify this in our revised paper. At the same time, as we state above, we acknowledge that because we do not provide data on these environmental changes in the manuscript, we reduce our emphasis on these points.

Second, one of the main focus points of the paper seems to be the difference in turbidity between the habitat types. However, rather than testing fish in different levels of turbidity they tested fish in two different light conditions. The authors do not provide any clear reasoning for doing so nor enough detail to understand how fish perceive the light conditions or tests if the 'glacial light' condition indeed resembles turbidity by providing less visibility. It is also not considered that besides a reduced visibility, the change in brightness and colour of the light may give the fish a different sense of safety and the behavioural responses observed may in that case be more to do with the risk-taking behaviour than with sensory input.

Response: The impaired visibility in glacial lakes is caused by increased turbidity, and characterized by steep changes in brightness as depth increases. We have measured these changes in irradiance using spectrophotometry in the lakes where our experimental fish were sampled. The light filters used in our experimental set-up were chosen to mimic these strong changes in light levels, and we used spectrophotometry data on a subset of lakes as a reference for this experiment. This is now clarified in the methods (from line 133) as well as in the new summary figure (Figure 1), where we present the irradiance spectra measured in a typical spring-fed and glacial lake, as well as the irradiance spectra

measured in our experimental tanks. We have now also modified our description of the light treatments throughout the manuscript to better emphasize the focus on the change in brightness and visibility occurring in glacial compared to marine and spring-fed environments.

Third, the experimental design was overly complex, with fish from four different habitat types tested with two light types and two odour cues. The authors thereby ran models with complex interactions and did a poor job in explaining the results, making it very hard to extract the key findings.

Response: We believe that the new version of this manuscript, following intensive rewriting of all sections, now allows the reader to understand our objectives, experimental design, results, and the meaning of these results in a much more straightforward fashion. The addition of Figure 1, graphically presenting the experimental design, will also greatly help the reader follow our study.

Fourth, the paper was really lengthy and far from to the point with for example the introduction being close to double the length of many ProcB papers. Also in general I found the authors did a poor job in explaining what they did and why, and in good order, such as stating experimental conditions before explaining the experimental design and very lengthy, hard to follow predictions without proper understanding of the experimental details. In combination with the complex interactions between the different factors and hard to follow results, the paper unfortunately does a poor job in explaining a potentially promising dataset.

Response: We acknowledge that clarity is important for making sure readers understand the paper's findings and implications. We have now completely re-written the introduction that we believe is much more focussed and directly highlights the context and goals of our study, as well as explaining what our design allows us to infer about evolution and plasticity of behaviour between populations. We have also shortened our predictions, which we believe are now much clearer. Throughout our revision we have kept the importance of clarity in mind, and endeavoured to make sure our meaning is lucid and important details are included - even within the space constraints imposed by the journal. The manuscript is now also somewhat shorter, and has been kept within the word limit for a paid version of a ProcB paper (Open Access), even with the addition of a graphical representation of our sampling sites and experimental design (Figure 1). We appreciate the more specific points below that helped guide us in this rewriting effort.

Lines 21-24: See my main point regarding the framing of the study in the context of environmental change.

Response: See our general reply above; both the abstract and the introduction now clearly frame our study in the context of newly formed glacial lakes, created by melting of glaciers now occurring at an unprecedented pace in arctic areas such as Iceland.

Line 24: Why novel sensory environments? Fish moving to different habitats will experience different environmental conditions yes, but the environments are not necessarily novel (sensory speaking).

Response: Although the glacial lakes we studied are indeed natural, many are formed recently because of the retreat of glaciers due to climate change, some within the past 50 years, thus indeed representing a novel sensory environment for sticklebacks colonizing them from the ocean, both in an evolutionary and in an individual sense. This has been clarified in the abstract and introduction.

Line 28: Strictly speaking the population's habitat did not affect anti-predator behaviour but was linked to it. Or in other words, you found that fish's anti-predator behaviour could be explained by their habitat.

Response: This sentence was edited to read: "Anti-predator reactions differed depending on the population's native habitat, [...]" (current lines 30-31).

Lines 70-73: A more complex genetic architecture does not necessarily mean evolution is slower; it could also be faster.

Response: We agree and have now removed this sentence from the introduction.

Lines 77-78: This is not the place yet to state what was studied and should be moved down or better, explained in a different way.

Response: This has now been completely re-written.

Lines 79-81: This is important information that should come way earlier in the introduction. Actually, it would make more sense to me to write the paper around this focus rather than starting with the climate change narrative, which is not so fitting to the actual study.

Response: We have followed the reviewer's advice and introduce the geographical and historical context of our study in the second paragraph of the manuscript.

Lines 86: What is meant with “current and historical” exactly? Make clear in the text.

Response: This has now been removed in the new version of the introduction.

Lines 86-87: If predation/predator evasion is suggested as a main potential selective pressure for sticklebacks in Iceland you should provide more detail about that here, such as the sympatry with predators and occurrence of which species. This is critical information currently missing from the manuscript (apart from a sentence in the discussion).

Response: Iceland’s freshwater fish predator species and their presence in the lakes we sampled are now introduced in current lines 85-86 of the introduction.

Lines 92-93: Again, information is provided about some part of the practical work before any real explanation of the study. Leave this for later in the introduction.

Response: This has now been moved in the rewriting of the introduction.

Line 100: burst swimming is not the same as a fast start and should not be confused with one another.

Response: The mention of “burst swimming” has now been removed.

Lines 103-122: This is a paragraph of 288 words meant to explain the study and the predictions. This paragraph is really way too long and is hard to follow. Furthermore, despite its length, the study itself is actually still not well described (only 2 lines) and much remains unclear. For example, did you conduct an experimental or observational study? Was it done in the lab? What is meant with “light environments” and is this experimental design appropriate for investigating responses to turbidity? Etc. State concisely what you did exactly, with what purpose, and what predictions. The predictions do not need the long explanations given here when the proper background information is given above.

Response: This paragraph has been completely rewritten, and split into two paragraphs. The first one, starting on current line 99, begins by explaining the experiments in detail and what our design allows us to infer, followed by shorter, clearer predictions (current lines 111-118), as suggested.

Lines 125-128: Here, the last three sentences of the introduction, the link is made again between turbidity and the potential relevance for human induced environmental change. This point is fine but is more for the end of the discussion, and framing the whole study in this context is not appropriate, see main point.

Response: The last sentence of this paragraph has been modified. See above also for more details about the reframing of the study and better justification of links to human induced environmental changes.

Line 133: You have to be more specific about where you caught the sticklebacks than “in Iceland”. Especially as you are testing fish from different types of sites and multiple replicates per site. I later found some of this information is provided in the SM. Integrate this and actually refer to that information.

Response: The paragraph on sampling procedures has been edited to refer to the map of sampled locations, now in our new Figure 1, as well as adding details about environmental measurements, elevation, and visibility for the different habitat types in the main text (starting current line 129).

Line 143: Clarify that tests were done with individual fish. How were these focal fish selected? How was the order of testing decided? Was population sufficiently randomized across trials? Much detail is missing about the experimental design.

Response: The paragraph introducing the trials was modified and clarifies that each fish was selected randomly and subjected to two trials (one per cue) in one of two light treatments (this is also reflected in the new Figure 1). Trial order was randomized while ensuring that for each population and light treatment combination, half of the fish were presented with the mechano-visual cue first and the other half were presented with the odour cue first. We aimed at measuring an equal number of fish in each light treatment for each population and now clearly indicate the mean and standard errors for the number of fish from each population tested in each light treatment and cue order combination in Figure 1.

Lines 147 onwards: Not sure why some words are underlined, but not a styling that ProCB supports as far as I know.

Response: This has been removed.

Lines 148-151: With what speed? Was this controlled and consistent across all trial? Was the replica already visible at the start of the trial or how was this prevented?

Response: The rubber trout was controlled by an Arduino program and a robotic arm, and the speed constant and consistent across all trials (0.77 m/s). It was covered by an opaque plastic tube before the start of the trial (i.e. during the acclimation phase and the other trial with olfactory cue). Details about this, previously placed in the supplementary

material because of lack of space, have now been added to the main methods (current lines 156-159).

Lines 157-161: Given that turbidity is one of your main interests and main focus points of the study it is unclear to me why you not use turbidity itself in your experimental design but rather use a difference in “light environment” instead. Furthermore, you need to explain a lot more about these treatments, such as why you used light environments, what do the light environments entail, how are they perceived by the fish, and if pilots have been done to test if they are a good proxy for testing of turbidity. Did you run any tests to measure underwater visibility and was that lower for the ‘glacial light’ environment? Has such an experimental paradigm used before at all?

Response: Difference in light intensity and corresponding changes in visibility is one of the most obvious parameters affected by turbidity in glacial lakes, and we chose to focus on this aspect in our simulation of spring-fed versus glacial lake environments for three main reasons. 1) Changing light environment provides a much more controlled, reliable, and consistent treatment than modifying turbidity directly in the tank. Weeks of pilot experiments taught us that adding powdered glacial milk to the water to increase total suspended solids makes for a rather “patchy” visual environment, as it tends to accumulate in certain places. One solution to this is to use continuous pumping and/or aeration in the tank, which was not possible for us as it would affect water movement and likely have confounding effects on our predator cue. Furthermore, adding glacial milk would have affected the water chemistry which likely would have interacted with the olfactory cue we used or the fish’s olfactory receptors, something we wanted to avoid as changing both water chemistry and visual environment together would decrease the interpretability of the results. If these problems could be overcome, it would certainly be an interesting follow-up experiment. 2) Additionally, we had very reliable measurements of light intensity for each lake sampled that was used to choose the light filters. The light environments in the experimental tanks were set up to mimic these natural measurements. This is now specified in the text and data on these light spectra, both measured from the lakes and in our experimental tanks, added with our new Figure 1. We think our experimental light treatments matches reasonably well the light conditions measured in nature. 3) Finally, modifying light environment made the experiment possible on a practical level because we could still see the fish. This was not possible in our pilot experiments where we tried to add powdered glacial milk.

Lines 168-170: This is unclear. So each trial a fish received a second cue? Why was that done? Very likely the first cue will influence the second cue, making it much harder to disentangle the effects than were they tested separately. Also, was made sure the order of the cues was randomized?

Response: Yes, this has now been clarified. There were constraints on how many individual fish we could catch during the short summers in Iceland. Therefore, a repeated-

measures design allowed us to increase our power with this limited number of fish by controlling for unmeasured and/or unknown factors that might affect inter-individual variability. Because trial order affects fish behaviour (see results and also response to Reviewer 2), the cue order was randomized amongst the fish such that half of the fish in a population received the odour cue first, and the other half received the mechano-visual cue first. Results are presented with the effect of trial order taken into account and the trial order effect is presented in the results. The experimental design is now clarified both in the text in the methods and in our new Figure 1.

Lines 175: What happened with the other 98 fish (~25% of the experimental fish)?

Response: Because of technical problems with the video recording, only a subset of the trials could be evaluated for reaction distance. All other measurements were scored live and thus not affected. This is now specified line 179.

Line 185: So each fish received two trials? This is not clearly stated anywhere before, neither how much time there was between trials, if that was the same for all fish, etc.

Response: In re-writing the description of the experiments, we are now introducing earlier (line 147) that each fish was subject to two consecutive trials (the two different cues) in one of two light treatments. The acclimation period both before, and in-between cues was the same for all fish each year, as was the measurement period, as stated lines 169-174.

Lines 186-187: You state interactions were included. But as you tested five fixed factors, what interactions did you actually include? And did you actually have some clear hypotheses for including all interactions?

Response: Interactions between habitat type, light treatment and predator cue were included in all initial models because they were important for testing our predictions. This is now specified on current line 194. We were specifically interested in the effect of the habitat fish came from on how they reacted to the two different cues in the two different light treatments. Interactions between habitat type and cue would suggest evolution of differential use of sensory cues; specifically, we predicted glacial fish might be more reliant on the odour cue than the mechano-visual cue and the reverse for spring-fed fish. Interactions between light environment and habitat would indicate plasticity in response to different cues depending on the current visual environment. A prevalent hypothesis in the literature is the “flexible stem” hypothesis which predicts reduced plasticity in derived populations. We in fact found the opposite and suggest that the reason glacial fish have the greatest plasticity could be due to the rapid changes in light environment that occur over short changes in depth in glacial-fed lakes. Three-way interactions between habitat, cue, and light environment are more complicated to interpret but were important to

include because they would suggest that the two-way interactions we were interested in were contingent on the third variable.

Results: You ran mixed models with a lot of fixed variables but explain your results seemingly focusing just on single factors. This is unclear, also how you extracted the results, i.e. the final model after removing non-significant effects or the full model, and how you accounted for other significant effects. For example, you are stating the effects of reaction time without saying anything about the effects of light condition. It should be much clearer how you acquired the different results and the individual results should be more clearly and concisely spelled out. Also, one normally starts with explaining the highest interactions and then explaining lower level interactions and then individual fixed effects as post-hoc tests, which is not the way it is done here. Finally, you seem to have run a lot of statistical tests but did not correct for this, such as by using a Bonferroni correction.

Response: The statistical analyses include 4 main models because we had 4 different response variables: one model looks at activity level, one at distance from the predator, one at the probability of freezing in reaction to a predator cue, and one looking at the probability of escaping with a fast start. This is now specified in the last paragraph of the methods. Corrections for multiple testing are typically used for models tested on the exact same dataset and response variable, which we have done for all contrasts done in this study (FDR correction for multiple testing using the R package emmeans). Adjusted p-values are reported in Table S6 for all contrasts.

We have now re-written the results completely, starting with a paragraph stating the main results from all four main mixed-models (specifically the complete results of an Analysis of Deviance for each model reporting the main and interaction effects). We then later describe particular aspects of the analyses that address our main hypotheses. We believe keeping the initial structure which first presents the entire model for each response variable and then presents more detailed results sorted according to our four main hypotheses, does help the reader make sense of the admittedly complex results of our analyses. Prior to our original submission, we tried the alternate approach of going through each model one-at-a-time, but it made the results with respect to our hypotheses extremely opaque and much more confusing than our current set-up. We have made extensive revisions to the results in this version to further increase clarity from previous version.

Lines 194-197: This should be part of the methods, not the results.

Response: The sentence has been moved to the methods.

Lines 199-200: This should already be stated in the introduction, not the results.

Response: This information has been moved to the introduction.

Lines 199-208: This paragraph is very hard to follow. Start by stating the results in such a way that it is clear what analysis you did, state the statistics, and then explain the result.

Response: This paragraph has now been completely re-written.

Lines 222-233: It is not clear if you presented the two types of cues in random order or if the olfactory cue always came second in the trial, which, if indeed the case, could just be an effect of timing.

Response: This has now been clarified in the methods. Cues were presented in random order and the order in which they were presented was included in all models, so the effect of timing is accounted for by the “trial order” fixed effect in our analyses.

Lines 227-228: Over what time period? Was this “after” period the same for both cue types?

Response: Yes, all trials consisted of an observation period of 8 minutes during which activity was scored, timed from when each respective cue was triggered by the push of a button. This has now been clarified in the methods and is repeated briefly again when results of activity measurements are first mentioned, current line 235.

Lines 228-231: hard to follow. Also a very specific result, taking two from the four population types in one light condition and looking at activity after the cue.

Response: This test, comparing differences in activity between cues in the light treatment most resembling the fish’s original habitat, was performed specifically to test the hypothesis that fish adapted to different light conditions may rely on different senses. The hypothesis is now briefly summarized again before introducing this specific result, for clarity.

Line 237: As commented on before, it is still completely unclear what spring-fed light and glacial light entails and what it means to the fish and how that is relevant as compared to the difference in turbidity the fish may actually be experiencing.

Response: Please see above our extensive explanation addressing this point, including why we focused on light environment rather than other variables affected by turbidity and how light environment in the experiment compares to what fish experience in the environment they were captured from.

Lines 242-247: Again a very specific result, and unclear. So you find a three-way interaction, but post-hoc tests actually showed only differences in activity existed for one habitat type? Also, what post-hoc test did you do here? This needs to be clear for each and every result you state.

Response: We have now clarified this result, the hypothesis it is testing, and the contrasts performed. The three-way interaction shows a complex relationship between habitat-type, cue, and light treatment. Here we are testing the hypothesis that fish from different habitats have different levels of light-environment-induced plasticity. Therefore, we test contrasts between light environments for each cue separately. In this case, highland glacial fish seem to be driving the difference in activity between glacial and spring-fed light environments and show more light-environment-induced plasticity than other populations.

Lines 250-251: What do you mean with 'differences between cues' and 'their native environments'? The light treatments used?

Response: This sentence has been removed for clarity.

Lines 256-269: Very complicated write-up again. Please make your results as clear as possible.

Response: This paragraph has now also been completely re-written in order to emphasize and clarify the results.

Lines 274-276: This detailed interpretation about evolutionary changes in use of sensory modalities and anti-predator behaviour does not follow naturally from the broader statement above that states just the existence of significant effects and interactions. This is more a statement for the concluding chapter. This initial paragraph would be a good place to provide a concise overview of the results, before going into detail discussing them.

Response: We have modified and added clarifying statements to this paragraph. We believe that the new version does give a concise summary of what we find to be the most salient points of an admittedly complex set of results. We believe further discussion of specific results belongs in separate paragraphs placing them in a broader context.

Lines 281-283: Rewrite as currently "these differences" and "this reduced activity" do not refer to anything. Recapitulate concisely what you found before making your interpretations.

Response: This paragraph was significantly rewritten and this sentence was removed.

Lines 283-285: A couple things. First make clear how fish were different rather than just stating they were 'more' different. Second, why would this be 'strong' evidence? Third, "evolved differences for these highland fish"? You want to say that as fish populations that were more distant from the founding populations were more different this provides evidence for evolved responses? Fourth, consider alternative explanations, such as could learning differences play a role or are there differences in how well the fish acclimatized to the lab/are used to disturbances.

Response: This paragraph was significantly rewritten.

Lines 290-291: As you didn't test this, I would stick to discussing your (many) results.

Response: This paragraph was significantly rewritten and we have tried to clarify our claims. We are in fact discussing our results by connecting them to biological phenomenon. In other words, we feel the discussion section is the place where the results should be interpreted in relation to potential evolved (or learned) behavioural strategies.

Lines 343-347: Is there any data on the genetic divergence on these or similar Icelandic stickleback populations or data that could inform the time it took for populations to establish the highland lakes?

Response: No, there is no current data, but this is an area of future research we are currently pursuing.

Lines 366-370: How do you account for any learned or plastic behavioural responses of the fish?

Response: We have tried to clarify the potential for learned or other forms of developmental plasticity, for example see the first paragraph of the discussion.

Lines 373-374: This sentence is not clear.

Response: We have revised this sentence.

Line 378: How do your behavioural results, linked to predator avoidance behaviour as you state yourself, help say something about colonization success? This is not directly clear.

Response: We believe that the substantial rewriting of both introduction and discussion make these links clearer. Sticklebacks are well known and studied for their extraordinary capacity to quickly colonize new habitats, and the young Icelandic glacial lakes we studied can only have been colonized recently. Changes in anti-predatory behaviour in response to different sensory environments is essential to survival, and better understanding the forces underlying these changes goes a step further towards understanding sticklebacks' capacity to quickly colonize and thrive in novel environments.

Referee: 2

In this manuscript the authors investigate how habitats varying in visibility, salinity and elevation affects anti-predator behaviour in three-spine sticklebacks. They test olfactory and mechano-visual predator cues on fish from four different habitats in two different light regimes over two years. They conclude that habitat did affect anti-predator behaviour and propose that their results suggest adaptive evolution and differences in behavioural plasticity in response to changes in visual environment.

In total, the authors tested almost 400 fish from 15 populations, and the manuscript indeed contains a lot of data (and hard work). The results are interesting both from a behavioural and evolutionary perspective, but also in a climate change context.

Response: Thank you!

I have a couple of questions and concerns. Firstly, given that the manuscript contains a lot of data, it is a bit difficult for the reader to keep track of everything. In order to fully understand the methods and results, the reader has to read the supplementary material, which shouldn't be needed. This is most likely a product of the journal style of only allowing a shorter methods section in the main paper, but the experiment should still be understandable from the main text. For example, the collection sites (Fig S1), sample size and population characteristics (Table S1) and full results from the statistical models (Table S2-S6) are all found in the supplementary material.

Response: We understand the need for some more details in the main paper in order to better clarify both our experimental design and statistical analyses. We now include a new Figure 1 that includes all collection sites, detailed information on the light characteristics of representative spring-fed and glacial-fed lakes and the experimental light treatments meant to emulate them, as well as the experimental design, including additional

information about sample sizes. In addition, the entire Analysis of Deviance model is given for each of our four models in the text at the beginning of the results section. Thus all main effects and statistically significant interaction effects are reported there. If a reader wishes to further see the individual parameter estimates for each fixed effect and significant interactions, these are still provided in the supplementary material as tables.

Along this line, the authors state on line 143 that they used a sample size range of 20-41 per population. But that was then split in two light treatments, and some fish did not meet the criteria for inclusion during the experiments and had to be excluded, leaving a sample size as low as 8 for some populations. The actual sample size should be included in the main paper, and potential concerns regarding the robustness of the results due to low sample size should preferably be mentioned in the discussion.

Response: We have tried to clarify sample size information by editing this part of the methods and adding information on sample sizes for each population x light treatment x cue type combination in the new Figure 1. While the sample size for a given population x light treatment x cue type combination is around ~12 (with ~6 per cue order), the analyses themselves are done not at the level of the population, but at the level of the habitat type. The sample sizes for a given habitat x light treatment x cue type combination is quite large, 27-63 (mean±se: 45.2±3.26) and we put this information in the figure legend for our new Figure 1. Detailed information about sample sizes can still be found in the supplement as well, now in its own table, Table S2. Thus, although we tried to anticipate the most likely questions about sample sizes and include that information in the main text, any other questions a reader might have about our sample sizes could be answered by going to this supplemental table.

The experiments were conducted in the summer, i.e. during the mating season. Although “non-gravid” adults were used (line 133), they were kept in the lab during 2-4 weeks prior to the experiments, which is ample time for sticklebacks to mature. Were the fish held in same-sex tanks during this time? If so, how did you ensure that they did not mate? When many males are held together during the mating season, they form dominance hierarchies with subordinate and dominant males exhibiting very different behavioural patterns. The sex of the test animals was confirmed by examining gonads, but sex is not included in the models, why? Given the likelihood that males and females differ in their behaviour, sex should at the very least be included in the models.

Response: Our fish were held in mixed-sex tanks, but visibly gravid females were avoided at the time of selecting fish for the experiment. The effect of sex was tested on each of our four main models but was not significant and did not influence our results, and was thus removed from the models in order to avoid convergence problems given the current number of fixed, interaction, and random effects already included in these models. All coefficients for the four models including sex as a covariates are now included in table S6 of the supplementary materials.

The authors also state that they checked presence of parasites (line 172), but nothing regarding parasite load is mention in the rest of the manuscript. Was there any difference in parasite load? If so that too should be included in the models. As with sex, parasites too can affect the behaviour of the fish, and sticklebacks can be very heavily parasitized. The authors themselves mention parasites as one of the factors that the fish needs to adapt to when colonizing new areas (line 63-66).

Response: Similar to sex, the presence of parasites did not significantly affect any of our results when included in the models. Note that we do not have information on parasite load, but on presence or absence of the main cavity parasite *Schistocephalus solidus*. All models including parasitism as a covariate are now presented in table S6. Note that while we did find a significant effect of parasitism on reaction distance, this seems to be driven by fish for which we did not actually have information on parasite presence. The effect of parasites on anti-predatory behaviour and use of sensory systems is the subject of a current follow-up study where we will dig more into details about how they might affect these behaviours. Because parasitism did not affect any of our main results and hypotheses tested in this manuscript, this variable was also removed from the models in order to reduce the overall number of variables. We now specify this lack of an effect of both sex and parasites in the results section.

On line 171, it says that brains were dissected for subsequent gene expression analyses, but this is not mentioned any further, no such analyses or results are included in the manuscript. Why is this mentioned when not included?

Response: We originally included this sentence because we felt it was a best practice to document everything that was done to the fish during the course of our experiment, but given this comment by yourself and the editor, we have removed it from the manuscript. The results of RNA sequencing are currently being analysed and are beyond the scope of this paper.

For the chemical alarm cue experiment, it seems that there was no control? I.e., the fish would usually react to the water movement itself, and in order to control for this, a control treatment adding/injecting water with no cues are typically used. This cannot be corrected at this point, but the authors should acknowledge the fact that their experimental design does not allow to disentangle the fish reaction to water movement only from the presence of CAC (or CAC+ predator cues, as was used in 2018), and what this might potentially mean for the predator avoidance aspect of their experiment.

Response: A combination of time and collection constraints during the very short summers in which fish could be collected and run through the experimental assay in Iceland meant that we had to make some trade-offs from the most ideal experimental design. One of these trade-offs was not including a control treatment injecting water only through the hoses used for the olfactory cue. This would have required having each fish go through an additional cue treatment and we know from this study and prior work that each subsequent treatment lowers their activity. With an additional treatment, this lowering of activity can create a floor effect in the data that is problematic. Pilot experiments performed in Michigan did not clearly indicate a consistent, strong reaction to water injected in the tank for any of the populations tested there. In 2017, we instead had the opposite problem, where many fish did not show any strong reaction to the olfactory cue, and numerous trials had to be aborted, hence motivating the addition of predator cues to the initial olfactory treatment. Furthermore, we do observe clear differences in reactions between our mechano-visual and olfactory cues, suggesting that fish react to more than water movement (which they also get with the mechano-visual cue) in these trials.

During the experiments, I assume altitude was the same for all fish? That would be the case if the university lab location corresponded to lowland. This means that while most habitat parameters were kept the same as where the population came from, all populations except for highland glacial were tested on the same altitude as where they came from. I.e., altitude was inevitably manipulated for the highland glacial population only? How do you think this might have affected the results?

Response: It is correct that the lab location corresponded to lowland altitude and that the highland populations were thus tested at an altitude different than that of their native habitat. The fish were however maintained at this altitude for several weeks prior to the start of the experiment, and we do not see any biological reason for altitude affecting the responses specifically measured in our experiment. Furthermore, differences were generally more pronounced between highland spring-fed and glacial lakes, suggesting that altitude *per se* does not explain divergence between habitat types.

There were many differences in experimental design between years. I understand that this was done because a lot of fish had to be excluded in 2017, and to make the fish more active in 2018 the authors added food cues during the experiment (it says in the supplementary material that: "In 2018, a transparent plastic bag containing three freshly thawed blood worms was attached to the end of the hose in the experimental tank to encourage fish to approach the location where the olfactory cue would be released"). They also added predator cues to the CAC cues, released the test fish directly into the experimental tank instead of using the chamber, used a different acclimation time etc. All of this information is found in the supplementary material but should be included in the main paper. Given the many differences in design between years, there is little surprise that year comes out as a strong factor in all models where it was included (Table S2-S5). This result is however

not mentioned in the results section or mentioned or discussed in the discussion section. The figures also display the data pooled over years, but this is incorrect. The authors need to present and discuss the year effect, and present their data split by year, as per their results.

Response: The inclusion of year effects in all models are now clearly stated in the statistical methods, and the significant year effect clearly mentioned on current lines 200-203. We agree that it is not surprising that we find a significant year effect in our models given these differences in design between the years. However, because this effect is accounted for in all models and all the results we discuss in this manuscript are robust to the inclusion of this year effect, these between-year differences do not in fact take away the differences we observe between habitat types, predator cues and light treatments. We believe that we are honest in our presentation of differences between years both in the methods and results and that our statistical accounting for the year effect is correct. Our results are indeed robust to these slight variations in design between the years.

Appendix B

Dear Editor and Reviewers,

Thank you for your feedback on our work and suggestions for improvement, please find below our detailed responses to each of the comments.

Associate Editor Board Member

Comments to Author:

The authors of this manuscript have done a good job at addressing the inquiries raised. The explanations are clear and help clarify previously confusing points.

Response: We thank you for recognizing our extensive revisions.

Several clarifications though should be included in the manuscript as concerns similar to those raised by the reviewers will likely come up among the readers.

Response: We acknowledge the importance of clarity to readers of our manuscript, and have worked hard to be more clear. In our previous submission, we included much more information relevant to reviewer queries in the response letter than in the manuscript. This was to keep our text within the journal word limit. We see that this frustrated the reviewers and makes our revisions less effective for future readers; therefore, in this current version, we have included additional text in the manuscript as suggested by reviewers below. We agree that it is generally true that if a reviewer has a question, then other readers likely will, too. We have tried to add information in a succinct way that will be informative but not distract from our main text. Note that because of differences in line numbers between word documents with or without track changes, all line numbers we are referring to below should this time match the pdf version of the manuscript.

The experimental design is now easier to follow and an extension of that is that additional queries and concerns have been raised about the study. In particular, one of the reviewers validly questions whether the experiment to address the effect of turbidity is indeed getting at that. The relevance of this study design needs to be stronger and better connected to the point at hand (turbidity) for a solid interpretation of the findings.

Response: We understand why our prior presentation of turbidity as a key difference between glacial and spring-fed lakes could be interpreted as the focus of our study, and have endeavored in our revision to clarify that there are other important differences between these lakes, which were the actual target of our experimental manipulations. Although turbidity is an obvious feature of glacial lakes, other aspects of the visual environment of these lakes such as brightness are likely critically important – and far easier to manipulate. In short, we agree that we are not addressing turbidity directly with this experiment. Previous work examining turbidity has either simulated it through growth of algae (which is not relevant here as turbidity in glacial lakes is

induced by silt from glacial melt, not algae), simulated it through introduction of silt with intense aeration (which has its own problems, including changes to water chemistry and creation of a very different water current environment from any the fish inhabit and potentially masking mechanosensory cues induced by predator attack), or instead used observational experiments conducted *in situ* (which removes the ability to experimentally disentangle genetic and plastic effects as we were attempting to do here). All of these experimental challenges get in the way of addressing turbidity directly. We made another pass through the paper to remove any language suggesting that we manipulated turbidity with the experimental design we used.

However, a related major difference between glacial lakes and all the other habitats in our study is their difference in overall light levels (or brightness, described in more quantitative detail in our new Supplemental Figure S1 and in Table S1), which is caused by the high turbidity of glacial lakes. We point out that the steep reduction in light levels that occurs in the glacial lakes is biologically relevant to animals such as stickleback fish that are highly reliant on vision; this is supported by other research investigating visual and behavioral adaptations to low-light environments, and provides a backdrop to our experimental design. On average half of the light that reaches 1m depth in spring fed lakes reaches the same distance in glacial lakes (new Supplemental Figure S1 and Table S1). Our experiment emulated this difference in brightness between glacial and spring-fed environments, and in our revision, we clarify that this was indeed our intent and provide justification. In brief, we used brightness data measured by spectrophotometry from representative highland glacial and spring-fed lakes to guide our experimental light treatment manipulations (detailed in Figure 1), checking the resulting experimental light spectra against field data to ensure we had a reasonable mimic of those differences while ensuring that we would be able to observe fish behavior under these manipulated conditions. This experimental manipulation resulted in at least a 74% reduction in light between treatments, severely reducing visibility, which we estimate would be equivalent to a difference in clear water in brightness as swimming from the surface to 6m depth. We have added these details to our revised text (lines 133-142 and 155-163).

An additional but also highly relevant point reflects concerns about the small effect size of the findings. While the results may be significant, the effect sizes should be clearly stated and fully discussed in the study.

Response: We agree in the spirit of this advice as there is certainly a difference between statistical and biological significance. In response to this request, we present marginal and conditional R^2 for all models in the main manuscript (results section), and effect sizes for all random and fixed effects in our supplemental material extracted in the form of semi-partial R^2 , using a recently available R package. However, it turns out that there is not much agreement in the statistical community on how to calculate effect sizes for complex mixed effects models such as ours, especially those based on non-Gaussian distributions. Therefore, we caution reviewers and readers to consider this uncertainty in their interpretations.

Extracting effect sizes from our models was not a trivial exercise, as methods to do this for generalized mixed effects models have only recently been developed. This task was made even harder by the presence of factors with multiple levels and interactions between them in most of our models. In the end, as we suspected, the effect sizes mirror what was already apparent from our statistical significance values. These estimates do provide additional information for interpreting our findings. Yet, it is an open question about how big an effect size needs to be to consider it “biologically relevant” and the literature we consulted makes it clear that the meaning of the effect sizes calculated from complex mixed effects models, in particular the relative proportion of the variance explained by specific variables when random effects are involved, can easily be mis-interpreted. The developers of the statistical package allowing estimation of semi-partial R^2 s for complex glmms warn that these are expected to explain small proportions of the total variance in the case of a Poisson distribution, due to a very large residual variance caused by inherent distribution-specific variance with such distributions. We thus present tables with semi-partial R^2 for all individual variables in the supplementary material, but also added a paragraph in the Supplementary Methods containing necessary caveats for the reader to judge on their own. Note that the total R^2 for our models (marginal and conditional, see results section), calculated both using the function `r.squaredGLMM` from the MuMIn and as the sum of all semi-partial R^2 extracted in “partR2”, explain a substantial proportion of the variance, particularly when compared to what is usually reported for behaviour and wild populations. These conditional and marginal R^2 estimates have now been added in the main text for each model.

Finally, we reject the notion that small effect sizes are irrelevant; for example, fitness differences of a few percent can have substantial evolutionary impact. We stand by our results, and readers now have all the necessary information to interpret the outputs of our models, including both significance and mean values from our initial analyses and the effect sizes in the supplementary material with necessary caveats. We also have additional information below addressing the reviewer’s specific comments on this issue.

Finally, reviewers provide excellent suggestions to improve the manuscript including rewording of the abstract and redesigning Figure 2 that is currently difficult to digest.

Response: We thank the reviewers for these suggestions, and they have led to substantial improvement once again of the manuscript. We have made extensive changes to the abstract and thank the reviewer for suggesting we do that. Although we disagree with the reviewer’s claim that pie charts are inappropriate for comparing categorical responses between groups, we do agree that Figure 2 was confusing and needed to be redesigned. We initially tried alternatives to pie charts for visualizations of the data, but found them to be even worse; for example, stacked proportion bar charts are sometimes used for these kinds of data, but we found that type of visualization was even more difficult to interpret. Note that the data we are trying to visualize are the proportion of fish reacting with each of four types of initial reactions, i.e. this is not a continuous response variable. Therefore, we have made an extensive revision of this figure that we think makes the main points more obvious. We placed the pie charts into a grid to facilitate comparison horizontally (comparing habitat types) and vertically (comparing light treatments)

and in two sets of plots (comparing olfactory & mechano-visual cues). We have color coded headings to use the color scheme we used in the other figures to denote different habitats and light treatments. We also include detailed quantitative values in supplementary tables (S12 and S13) with the raw numbers of fish using each reaction in all combinations of habitat types and light treatment for each of the cues.

Reviewer(s)' Comments to Author:

Referee: 2

Comments to the Author(s).

Review of the revised version of manuscript RSPB-2021-1881. The authors have made a great effort to revise the manuscript and it is now much clearer regarding many of my previous comments. In particular the inclusion of new Figure 1 makes the methods and sample size readily available to the reader.

Response: We thank you for noticing the large effort we put into our previous revision and hope you will notice the additional large effort put into our newest version. We also are happy to read your positive comments about Figure 1 and agree about its utility in demonstrating the sampling scheme, experimental design, and sample sizes to the reader in an easy-to-digest format.

For some of my previous comments, the authors provided a response but did not add or change anything in the manuscript, although in some instances the information provided in the response needs to be included in the manuscript. See my specific comments regarding this below.

Response: Thank you for the specific suggestions you make below for information we should have included in our last revision; we have implemented them.

In particular, I find it odd that the authors almost completely ignore a significant effect because they do not concern it to be of interest to the study (line 540). In their response the authors say that the between-year differences do not affect the results presented regarding habitat types, predator cues and light treatments. But how can the authors know this? The potential interactions between year and any other fixed effect are never tested (see my last comment).

Response: We did not intend to ignore a significant effect at all, in fact it was expected given the changes we made to the experiment to improve participation of the fish, which is desirable for meaningful research. We have a very detailed response about the specific comments made by the reviewer further below.

Note that the line numbers below refer to line numbers in the word manuscript file with track changes, and not the proof pdf.

Line 231: Predation can be a strong selective pressure for wild fish irrespective of their size, as long as large enough predators are around! I'd hence remove the word "small" from this sentence.

Response: Done; thank you for the suggestion.

Line 233-235: It would be appropriate to cite some of Ulrika Candolins work here, since they have shown that stickleback do rely on other cues (olfactory) than visual under turbid conditions. This was mainly tested in a sexual selection context, but is valid here and supports your expectation. Candolin 2009 provides an overview, specifically see the section in that paper with the subheading "switch to other cues"

*Candolin U. 2009. Population responses to anthropogenic disturbance: lessons from three-spined sticklebacks *Gasterosteus aculeatus* in eutrophic habitats. J. Fish Biol.*

Response: Thank you for this great suggestion. We have added this reference where suggested as well as in a relevant sentence in the discussion (now lines 88 and 385).

Line 288-290: I previously asked the authors regarding housing of fish prior to the experiments, and the effect of sex, given that the experiments were performed during summer, i.e., the stickleback's mating season. The authors did provide a response but as far as I can tell, some of the information was not added to the manuscript. The authors need to add to the manuscript that the fish were kept in mixed-sex tanks and that gravid females were avoided when selecting fish for the experiments (as per their response to my previous comment). The authors did not respond to my comment regarding dominance hierarchies in males, something that occur when keeping stickleback males together during the mating season. Since it has been added to the revised version of the manuscript that the authors did not make any specific selection regarding what fish to use in the experiment (line 298 states that fish were selected haphazardly, which isn't completely true for females then, since the authors responded that gravid females were avoided), I assume that the use of dominant and subordinate males in the different treatment groups would have been at random.

I suggest adding the information regarding mixed-sex tanks to line 290, for example like this: "and housed 1-4 weeks in 13 litre opaque tanks by population (i.e., in mixed-sex tanks)"

I suggest that the authors add to line 298 that gravid females were avoided:

"Each fish, selected haphazardly from its holding tank (note however that gravid females were avoided)"

Response: Thanks for pointing out the need for clarity in the manuscript itself. The way in which we held the fish did not allow for mating (which requires substrate for nesting). Our extensive

prior work on mate choice (and the many fish we housed for that work) revealed to us that the absence of suitable nesting sites and materials also reduces the competitive interactions and establishment of dominance among males. Due to our housing conditions and our haphazard selection of fish we are confident that we did not use all dominant (or all subordinate) males. To clarify these points in the text, we added line 144 “mixed-sex” as suggested to the text as follows: “...and housed 2-4 weeks in 13 litre opaque mixed-sex tanks by population...”. We made your other suggested change as well line 152: “Each fish was selected haphazardly from its holding tank (gravid females were avoided), [...]”.

Line 289: Add that the university laboratory was at a lowland location. The authors responded to my previous comment regarding the fact that all populations except for highland glacial were tested on the same altitude as where they came from (i.e., meaning that altitude was inevitably manipulated for the highland glacial population only), but they did not add anything to the manuscript. At the very least, the information that the laboratory was at a lowland location should be.

Response: We realize we did not address this point, so we now say line 143: “...non-gravid adult fish were transported to Hólar University’s aquatic laboratory (Verið in Sauðárkrókur, a lowland coastal town) ...”. Elevation in our study is a proxy for distance to the ocean and therefore likely age of the population (see lines 189-190); we are not studying altitude effects *per se* and not expecting them to interact with any of our treatments. We would also like to emphasize that there was a 2-4 weeks acclimation period before testing that we describe in the text.

Similarly to the above, the authors responded to my comment regarding lack of control (i.e., only water) in the CAC and CAC+predator cue treatments, but they did not add anything to the manuscript. I think their line of reasoning is OK, but it should be provided to the readers and not only to reviewers. Given the space limitations in the main paper I would add this to the supplementary material, where their pilot trials and the other aspects of not including a control can be mentioned.

Response: Thanks for pointing this out. We have added information provided in the previous response letter to the supplementary methods: “A combination of time and collection constraints during the very short summers in which fish could be collected and run through the experimental assay in Iceland created trade-offs between time available and the most ideal experimental design. One of these trade-offs was not including a control treatment injecting water only through the hoses used for the olfactory cue. This would have required having each fish go through an additional cue treatment and we know from this study and prior work that each subsequent treatment lowers stickleback activity. With an additional treatment, this lowering of activity could create a floor effect in the data that is problematic. Pilot experiments performed in Michigan did not clearly indicate a consistent, strong reaction to water injected in the tank for any of the populations tested there. Furthermore, we do observe clear differences in reactions between our mechano-visual and olfactory cues (see

Results), suggesting that fish react to more than water movement (which they also get with the mechano-visual cue) in these trials.”

Regarding my comment on the wear effect, the authors have now clarified this in the statistical analyses section of the main paper that year had an effect, and further write that “but are not of specific interest for hypotheses tested in this study” (line 538-540). While it is good that the authors at least mention the year effect in the main paper (although it would have been better suited to do so in the result section rather than in the statistical analyses section), it is incorrect to almost completely ignore an effect because the authors do not concern it to be of interest to the study. In their response the authors say that “year” is accounted for because it is included in the model, and that the between-year differences do not take away the results presented regarding habitat types, predator cues and light treatments. But how can the authors know this? As far as I can tell from the models (Tables S4, S5, and all of table S7), the interaction between year and any other fixed effect is never tested.

Response: There are a number of issues that you bring up here that we would like to address.

First, as we state in the Methods, several changes were made to our experimental to increase participation of fish in the second year. Specifically, we expected activity level to increase on average from 2017 to 2018 due to these changes as this was our intent when making those changes. This is in fact what we saw, indicated by the significant year effect, which we model as a difference in intercept between the two years (a fixed effect) to account for this increased activity due to our successful methodological changes. So, we do not mean to imply that the year effect is of no importance whatsoever (and we changed our text lines 212-215 to clarify this), but more so that we expected a fixed effect of year to be driven primarily by methodological changes rather than biological effects. It is of course possible that there were on average differences between populations between years for biological reasons, and this possibility is to some extent also accommodated by including the main effect of year in the models.

Second, you point to an interest in the interactions between year and the factors that motivated our experiment and for which we had specific hypotheses and predictions (predator cue, light environment, habitat origin). We had tried this early on in our analyses. Unfortunately, including these interactions in the model adds 7 parameters to an already complex repeated-measures mixed effects design, and the models simply do not converge. Even if they did, these extra parameters would use a significant number of additional degrees of freedom, reducing statistical power. Therefore, we think this is the best we can do given our data and sample sizes (even as large as they are). In fact, this approach means we will only detect effects that were consistent across years (our next point below).

Note: Details on parameters required to estimate all interactions with year: 1) a year x predator cue, 2) a year x light environment, 3) a year x habitat origin, 4) a year x predator cue x light environment, 5) a year x predator cue x habitat origin, 6) a year x habitat x light environment, and 7) a year x predator cue x habitat x light environment.

Third, we think that we are being conservative in our estimate of effects. This is because if these year interactions were in fact significant, the ones that might attract the most attention of a reader are those reflecting changes in sign or at least large changes in slope/direction. However, as there are only two years, positive and negative slopes would cancel out in our current analytical approach and we would detect no effect. This means that if we are missing anything, it is only those effects that are inconsistent across the two years of our study. That we DO find some effects suggest that those effects are robust to year differences and our methodological changes.

We made various edits in the Methods (lines 210-218) to address these points, and hope we have provided greater clarity and justification for our analytic approach.

Referee: 1

Comments to the Author(s).

I previously reviewed this manuscript and think the authors responded well to my and the other reviewer's concerns and much improved their manuscript based on our feedback. The considerably revised text is more focused and reads well. The authors also better now provide more background information and explanation regarding the light treatments and better re-framed the study.

Response: We are happy you noticed our extensive efforts and that they were helpful in clarifying the questions of yourself and the other reviewer.

I still however have two main concerns as well as some other comments that should be addressed. My first main concern is that what you are doing with your light treatment is simply comparing fish's anti-predator behaviour at normal light versus slightly dimmed light. Although higher turbidity goes together with faster declines in brightness, only using a difference in light intensity (and color) is not the same as a difference in nr of suspended particles. The aquatic environment may be darker but will have a similar level of transparency. Furthermore, looking at Figure S1 the difference in brightness between the treatments actually seems quite minor. I am therefore still questioning a bit how relevant your study design is to look at anti-predator behaviour in the context of the differences between the habitats in the wild. Also, the secchi distances you document for the glacial lakes were 0.8, 1.26 and 1.6m, which is considerably more than the response distances the fish showed to the fake predator.

Response: We have endeavored in this revision to make clear that we are (quite drastically) changing light levels, not turbidity. We have made multiple edits to this effect as we mentioned to the associate editor in detail above. We hope our new edits will allow readers to focus on what our experiment does test and why we did this.

In this spirit, we would like to make a few points here.

First, a general point. While we understand and agree with the point that there is a difference between decreasing light levels and scattering and/or occluding light with suspended particles, we maintain that there is nothing biologically trivial about decreases in light levels. We offer a simple thought experiment. Imagine wandering through a room cluttered with toys from a toddler with the lights fully on. Now imagine trying to navigate that room with the lights dimmed to levels similar to twilight. The transparency of the air remains the same and the major change is the amount of light (or brightness). Would navigating the room be as simple? Would you be more likely to run into obstacles or trip on something? Simply put, “dimmed light” is biologically meaningful to many animals, including to stickleback fish which are highly reliant on vision for finding food and evading predation.

Second, an experimental design point. The decrease in light levels between our two light treatments is not minor. The difference can be quantified by comparing the area under the curves in Figure 1c between the two treatments (integrating this area is a standard measure of brightness in visual studies); this results in a 74% reduction in brightness, which is equivalent to a 6m difference in depth in the spring-fed lake – a considerable difference in depth for a fish that is only several centimeters long. Thus, we did not make a minor change between our two light treatments, but a very substantial one.

Third, former Figure S1 (now Figure S2 in the revised version of the supplementary materials) is not an appropriate figure to use to quantitatively compare the light levels of the two treatments, and is instead included to give an overview of what the experimental tanks look like. This is because the camcorder attempts to adjust for lower light levels (indeed it is engineered to enhance visibility in the image under low light); therefore, the qualitative differences in light levels one can see in that figure are much less than what those quantitative differences were in reality. Instead, at lower light levels, the camcorder imaging results in increased pixelation as the differences in light captured by neighboring pixels of the digital sensor goes down and the background noise of that sensor becomes more apparent. This increased pixelation can be seen, for example, in the less crisp gridlines and in the increased fuzziness of the robotic trout in the image from the ‘glacial’ light treatment compared to the ‘spring-fed’ light treatment as well as the other image artifacts one can see in that picture. This feature of digital camcorders was extremely helpful in our ability to observe the fishes’ behavior (indeed it was a motivating reason we used these specific camcorders and scored through the transmitted image on a screen rather than direct observation) – we could still watch the fish with high accuracy. However, it does give the misleading impression that light levels are higher in dimmed light than they really are without this ‘camcorder correction’ which of course, the fish would not see. We added text addressing this point to current Figure S2’s legend as we are sure this information could be helpful to other readers.

Fourth, the secchi disc measurements are not wholly relevant to this particular issue. We did not intend that the robotic predator should be invisible to the fish. Rather, we intended that information through the visual modality would be impaired in a way that mimics reduced brightness in glacial lakes. Secchi measurements were originally included to show visibility differences, which we have now removed because of a separate issue this reviewer pointed out

to us below. We have now calculated the proportion of surface light that reaches 1m depth as a metric (measured in most lakes using spectrophotometry, now in the manuscript current lines 136-142, as well as in Supplementary Figure S1 and Supplementary Table S1).

We took these criticisms to heart and responded sincerely, by making changes to the manuscript and adding additional data and analyses to the supplementary material in order to make our methods more obvious. We hope that we have alleviated your concerns on this matter as much as possible, given the constraints we (and others) face of running behavioral experiments with wild animals.

My second main concern is that your effect sizes seem rather weak. In your manuscript you mostly focus on the existence of significant effects, but the stated differences in the results and those visible in the figures suggest the effects are very limited, with huge overlap between the fish from the different habitat. This questions a bit how important the effects are that you are describing and at least you should better state the effect sizes in your results and discuss their relevance in your discussion.

Response: We can see the relevance of effect sizes to interpreting our findings, and in other work we have incorporated effect sizes for these reasons. In response to your request, we have added effect sizes to supplemental material and addressed this point above. We also point out in the Supplementary methods the fact that for models such as ours there is no consensus on how to calculate effect sizes, because effect size estimates can be misleading for models that use Poisson or Binomial distributions, with additional complicating elements such as multi-level factors and multiple interactions. Moreover, in our opinion, the effect sizes support our claims initially derived from the original analyses.

We reiterate our earlier response to the associate editor about this point here to be comprehensive:

“We agree in the spirit of this advice as there is certainly a difference between statistical and biological significance. In response to this request, we present marginal and conditional R^2 for all models in the main manuscript, and effect sizes for all random and fixed effects in our supplemental material extracted in the form of semi-partial R^2 , using a recently available R package. However, it turns out that there is not much agreement in the statistical community on how to calculate effect sizes for complex mixed effects models such as ours, especially those based on non-Gaussian distributions, nor agreement on how reliable these estimates are. Therefore, we caution reviewers and readers to consider this uncertainty in their interpretations.

This was not a trivial exercise; to get this information, we worked for many hours to calculate these effect sizes. This task was made even harder by the presence of factors with multiple levels and interactions between them in most of our models. In the end, as we suspected, the effect sizes mirror what was already apparent from our statistical significance values. These estimates do provide additional information for interpreting our findings. Yet, it is an open question about how big an effect size needs to be to consider it “biologically relevant” and the literature we

consulted makes it clear the meaning of the effect sizes calculated from complex mixed effects models, in particular the relative proportion of the variance explained by specific variables when random effects are involved, can easily be mis-interpreted. The developers of the statistical package allowing estimation of semi-partial R^2 s for complex glmm's warn that these are expected to explain small proportions of the total variance in the case of a Poisson distribution, due to a very large residual variance caused by inherent distribution-specific variance with such distributions. We thus present tables with semi-partial R^2 for all individual variables in the supplementary material, but also added a paragraph in the Supplementary Methods containing necessary caveats for the reader to judge on their own. Note that the total R^2 for our models, calculated both using the function `r.squaredGLMM` from the MuMIn and as the sum of all semi-partial R^2 extracted in "partR2", explain a substantial proportion of the variance, particularly when compared to what is usually reported for behaviour and wild populations. These conditional and marginal R^2 estimates have now been added in the main text for each model.

Finally, we reject the notion that small effect sizes are irrelevant; for example, fitness differences of a few percent can have substantial evolutionary impact. We stand by our results, and readers now have all the necessary information to interpret the outputs of our models, including both significance and mean values from our initial analyses and the effect sizes in the supplementary material with necessary caveats."

We will add that the effect sizes we estimate for our behavioral data are very much in line with others in the behavioral literature, and seem reasonable considering the typically high variability of behavior, especially in wild animals, and given the various factors and random effects we model statistically. We do not claim that they are high, but do think that they are substantial enough to suggest they could be biologically meaningful.

Other comments:

- Abstract: You should revise your abstract to more clearly spell out the key results. Currently, you talk about "reactions difered" and "fish from different habitats difered in their reactions" and then immideately go to the interpretations, which thereore do not naturally follow. You also end with four sentences in which you are basically restating what you said in the sentences before, valuable space you could use there to state the specific results. This is especially needed because of the complex experimenta ldesign used.

Response: We significantly rewrote the Abstract in response to your suggestions, and hope we have clarified our findings and avoided repetition.

- You have data of two years but didn't clarify how your 15 waterbodies were allocated across the two years. Did you for example get fish from all 15 waterbodies in each year or 7 in one year and 8 from the other? If more the latter, was made sure that habitats were equally sampled across both years?

Response: All habitat types were represented in both sampled years as evenly as logistically possible. Seven populations were sampled in 2017, including two highland glacial, two highland spring-fed, one lowland spring-fed and two marine populations. In 2018, nine populations were sampled, including one highland glacial, two highland spring-fed, four lowland spring-fed and two marine populations. One of the highland spring-fed populations was sampled both years. This information has now been added to the Supplementary Table S2.

- *Lines 111-112: You state to expect the biggest differences between spring fed and glacial habitats. As those are the only two habitats you tested it is unclear what you are trying to say here.*

Response: We also tested marine fish, and these stand in for the ancestral state (all freshwater populations we examined descend from marines). Our statement reflects that we anticipate high divergence between freshwater habitats as they evolve away from their ancestral values. This is now clarified for prediction 1 in the main text (current lines 111-113).

- *Lines 131-137: you calculate percent visibility by dividing secchi depth by local lake depth. This is wrong as it would result in a deeper lake having a lower visibility. Looking at the supplementary I understand you tried this approach to account for many of the lakes being too shallow to lose sight of the secchi disk. Therefore an alternative approach would have been preferred to measure water transparency. Since that is not possible anymore without going out again and remeasuring, I suggest to just use a categorical variable.*

Response: We understand your point here, and have replaced the secchi depth derived percent visibility measure with a quantitative measure based on spectrophotometric measurements at different depths (new Supplemental Figure 1 and Supplemental Table 1 with some text replaced in the manuscript lines 136-142). Our new measure is the same that we used in Figure 1b to compare the spring-fed and glacial lakes used to guide our experimental light treatments. Briefly, we integrated the area under curves of light readings of downwelling light at 1m to measure the total amount of light at that depth. We then divided that by the amount of light at the surface, to get a measure of what proportion of light at the surface reaches 1m depth at the different sampling locations. Using these relative values (surface vs 1m) 'corrects' for differences in weather and time of day that might influence the amount and quality of light available at the time of sampling, but should capture the effect of the water on how much of the light is transmitting to that depth. And again, we are trying to estimate brightness differences to capture the effect on overall visibility – we are not trying to imitate turbidity *per se*.

- *Line 169: "Live scoring". How many unique observers contributed to this and were they naive in terms of knowing the habitat the fish came from?*

Response: There were 8 unique observers across live scoring trials, and one unique observer scoring reaction distance from videos. The observer identity was included as a random effect in

all our statistical models. For logistical reasons, it was impossible for the observers to be naïve to which habitat fish came from. This is because the water in the tank had to be adjusted to the natural conditions the fish came from, for each fish. For example, sea water was used for marine populations, and the temperature of holding and experimental tanks were adjusted to match that of the location of origin. The observers were however naïve to the expectations of this experiment, and we did extensive training to ensure high inter-observer agreement. Note also that because trials were run in parallel, several different observers ran trials for each of the populations and experimental combinations, minimizing the risks for a strong bias introduced by any one observer on population-level or factor-level estimates. The number of observers (as well as number of individual fish and populations, the two other random effects in our models) has now been added to the legends of Supplementary Tables S3-S5 with the detailed outputs of our statistical models.

- *Line 228: a slower response not necessarily means they are slower at detecting. Rephrase.*

Response: We have the qualifier ‘might’ in that sentence, to indicate this is but one possibility and we acknowledge that the behavioral response does not equal the perceptual difference, so we changed the word ‘detect’ to ‘react’.

- *Figure 1. For B and C, why do you only show data for two of the lakes? In C it remains unclear what the different lines are, i.e. what depths they correspond to. From C it appears that the Glacial light treatment does not correspond well to the irradiance measured in the glacier lake. With D, I appreciate your attempt to clarify your study design but this figure doesn't really add much and would not have that in the main text. Also note that your fontsize is far from consistent across the panels a-d and that the labels are far too large.*

Response: As stated in the legend, data are shown for the two lakes on which we based our light treatments. Our goal by presenting just these two lakes in the main figure is to be as transparent as possible about our methods. We have made some additional edits to the figure legend to increase clarity.

Given constraints of field work in both time commitments and travel to these often remote lakes, along with the fairly substantial challenges of getting boats onto lakes that are several km from roads which is required in order to collect spectral data (quite a bit of backpacking and carrying heavy loads of sensitive equipment and boats is involved!), we did not have the luxury to measure all lakes before conducting the behavioral experiments. Therefore, we initially sampled two lakes that were judged to be representative of the two habitat types by multiple local Icelandic scientists knowledgeable about Icelandic lakes. We now present data from other water bodies used in our study in the supplement (Supplemental Figure S1 and Table S1) and this data shows that these two lakes were not outliers for their habitat type (confirming what Icelandic scientists had told us).

We changed font size to try to make it more aesthetically pleasing and to make it clearer what depths are referenced in part C, and hope we have succeeded here.

Our intent was to manipulate brightness rather than the full light spectrum or actual turbidity, so we did not exactly match the specific shape of those curves. We do think we came fairly close, though, especially compared to similar studies that manipulate light environments.

We disagree with removing part D given that reviewers found our experimental design to be complex. We think it will be useful to readers for following the logic and design of our experiment, and we personally appreciate papers with similar figures.

- Figure 2 is hard to comprehend. Readers should within seconds be able to see the main result from the figure. Now the reader has to first read the axes, then the labels, then go over the pie charts, then look at the legend, to then go back to the pie charts to start and see how they are different. Pie charts are not ideal for comparisons nor show differences in the spread of the data.

Response: We agree this figure could be clearer and have made extensive revisions to that end in response to your suggestions. While we do think pie charts are often used inappropriately, in this case, they allow a very quick summary of the relative differences in the first behavior done in response to the predator cue. Other methods for this (e.g., stacked bar charts) are, in our opinion, much worse. We did generate stacked bar charts for our data and found that the results were pretty opaque using this visualization. We consulted with others who agreed with us. Note that these charts represent a proportion of fish reacting with each of four types of initial reactions, i.e. this is not a continuous response.

Therefore, what we did to improve readability and visual information content is to make the figure into a grid to facilitate comparison horizontally (comparing habitat types) and vertically (comparing light treatments) and in two sets of plots (comparing olfactory & mechano-visual cues). We also include detailed and quantitative values in Supplementary Tables (S12 and S13) with the raw numbers of fish using each reaction in all combinations of habitat types and light treatment for each of the cues. We have color coded headings to use the color scheme we used in the other figures to denote different habitats and light treatments.

In sum, we have worked hard to respond to the comments of both reviewers and the associate editor, and think that the manuscript has improved substantially from this process. We appreciate your input and the chance to revise, and sincerely hope that our efforts have resolved concerns sufficiently. We think that our paper can make an important contribution to the understanding of behavioral anti-predator responses in environments that differ substantially in visibility; and because such ecological changes are occurring at a rapid rate and are increasingly common in today's world, our work has broad relevance making it highly suitable for the journal.

Appendix C

Associate Editor Board Member

Comments to Author:

The manuscript has improved tremendously in clarity and I comment the authors for their work revising this work. This is an interesting study that uses an impressive data set. However, the experimental design does not allow the authors to tease apart whether the changes they are seeing in anti-predatory responses are due to the effect of early experience and/or genetic changes/adaptations. As only adults were tested, early experience could explain the differences in antipredatory behavior found here. The conclusions about rapid adaptation in antipredatory strategies is thus premature in this system.

The manuscript takes big leaps in concluding that rapid adaptation is at place without solid evidence this is the case. While I see the charm associated with rapid adaptation, acknowledgment of the limitations of the study is necessary. Given that the study is framed around rapid environmental changes, the role of early experience is also interesting. Even with this limitation, the study still provides valuable insights that will be of interested of ecologists and evolutionary biologist.

We thank the Associate Editor and Reviewer 2 for the recognition of our efforts in improving this manuscript. We certainly think it has come a long way and it is much stronger due to the detailed feedback we have received from the Reviewers and Associate Editor and are grateful for their time and effort.

We appreciate the Associate Editor pointing out developmental plasticity as a non-mutually-exclusive alternative explanation for our results. While we previously mentioned this possibility in one place, we agree with the Associate Editor that we were pushing the adaptive evolution explanation harder than warranted by our experimental design and have made edits in multiple locations to correct this (specifics below).

Many of the comments were about our Discussion and one of those comments made us realize that we needed to improve our communication about one of the key motivations behind our study design, specifically why we sampled lakes from different elevations (lowland vs. highland). We did some reorganization of the Discussion as a result and some additional edits that are described further below. We hope that this reorganization and editing increases clarity, reduces repetition, and more fairly presents what we can and cannot know from our study design. We thank the Associate Editor again for this guidance that we believe has made this paper better once again.

The full data and code necessary to reproduce our results have now been uploaded to Dryad. This material will be made public upon acceptance of the manuscript, and is currently available at the following link for review:

<https://datadryad.org/stash/share/yE9y3i6klryYYZWGCM5pfmN2oXO5a6OJgk1bz31Dfcs>

Specific comments:

Abstract: conclusions of rapid adaptation need to be tone down.

The last sentence of the abstract has been modified as follows: "This study, leveraging natural, repeated invasions of novel sensory habitats, 1) illustrates the rapid changes in antipredator behaviour that follow due to adaptation, early life experience, or both, and 2) suggests an additional role for

behavioural plasticity enabling population persistence in the face of frequent changes in environmental conditions.”

Line 43: change to “habitats are rapidly changing because of”

Done

Line 67: what do you mean by “these issues”? be more specific.

This has now been clarified; the sentence now reads: “Here we seek to understand how colonization of highly turbid versus clear lakes affects antipredator responses, focusing on differential use of senses.”

Line 71: add a citation to support the claim of adaptations in physiology

Done

Line 76: provide a reference value for “higher elevation”, >400masl

Done

Line 104: rather than calling them “alternative”, it would be more straightforward to call this the “novel visual environment”

Done

Lines 111-113: it would be helpful for the reader to add the rationale behind the first prediction, like you did on prediction 3. While the reader may be able to guess the thought process that took you to such predictions, for the sake of clarity it would be better to flesh it out.

Thanks for pointing this out. We added the rationale behind prediction 1, that now reads: “1) Fish differ in their overall reaction to predators depending on their native habitat, with the biggest differences expected between spring-fed and glacial habitats, as compared to putatively ancestral marine populations. This is because we expect that fish in spring-fed and glacial habitats evolved in different directions from the marine ancestor as they adapted to environments with very different visibility (Figure S1).”

Line 114, prediction 2, do you mean the magnitude of the difference depends on their native sensory environment?

Yes, this has been clarified.

Line 161-162, change to: "...an equivalent reduction in light as the one experienced by as fish in a spring-fed lake..."

Done

Line 208, what predictions were those? It is unclear what the authors mean by 'below' and how those predictions are different from the ones in the introduction.

This was a typo; we have removed the text "described below". The predictions referred to here were not different from the ones mentioned in the Introduction.

Line 304, even though the sample size is impressive, and I understand the authors are proud of it, the number of fish and populations have already been highlighted before. I suggest de-removing it from here and focusing on the conceptual gain of the study.

Good point. We removed the number of fish and populations from the sentence; it now reads "We show that habitat of origin, predator cue, and light environment all affect threespine stickleback anti-predator behaviour, often in interacting ways."

Lines 310-311, incomplete sentence. Needs to be re-written, here is a suggestion I think encapsulates the idea of the authors: "our comparison addresses lakes at different elevations and thus involves populations separated over many generations suggesting some evolutionary change". However, this is not a convincing argument for evolutionary change. Even though it seems likely, the authors cannot rule out the role of early experience. Indeed, there could be a complex interaction between experience and genetic differences among populations shaping the findings shown here. The authors should more transparently discuss the speculative nature of the conclusion of evolutionary changes specific to antipredatory behavior.

We appreciate the suggested edits. The suggested sentence didn't quite capture the point we are trying to make regarding elevation and so made us realize that we needed to clarify this point further. To that end, we have reorganized the Discussion by moving around several paragraphs and added additional text to clarify how we interpret differences in behaviour between spring-fed lakes with the same visual environment but different putative levels of genetic divergence from marine fish. We also acknowledge the potential role of developmental plasticity and the need for further experiments, for example, including reciprocal transplants between habitat types, in order to disentangle early environmental effects, genetic effects, and their interaction.

Lines 331-333. Rapid evolution may be happening but further work that confirms this hypothesis is necessary as the current study does not scaffold the effect of experience vs genetic responses. This sentence needs to put in context. A common garden experimental design would be necessary to arise to the conclusions posed here.

The end of this paragraph, now lines 340-346 was edited as follows to acknowledge the need for a common garden experiment to tease apart early life effects from genetics: "When we compared highland fish reactions in their simulated native visual environments, we saw strong divergence in preferred sensory modality further suggesting a match between native visual environment and

sensory cue use expected with local adaptation or developmental plasticity. Given how young Icelandic highland lakes are, the differences between the highland lake habitats and the lowland and marine habitats must have occurred very rapidly. Future experiments using common-garden experiments will be necessary to disentangle any potential effect of early life experience from genetic differences.”

Lines 373-375, these results are also in line with the idea that experience may be shaping such antipredatory behaviors as the highland fish seem to live in low predation environments.

We do not have data to speak to whether highland fish experience less predation than lowland fish. There are certainly piscivorous fish in those lakes. In various other places in the manuscript we addressed your points to clarify that experience/developmental plasticity could also be playing an important role. Here, we are trying to emphasize the benefit of comparing habitats at different elevation but with similar sensory environments to look at how divergence time (whether populations that have little versus more gene flow with marines) affects the magnitude of change more so than whether this is evolutionary or plastic change. We now open the paragraph, current lines 347-349, with the following sentence: “While our current study design makes it difficult to tease apart the relative influence of local adaptation and developmental plasticity, we can make some inferences about the role of genetic divergence on behavioural changes due to how geography constrains gene flow in Icelandic stickleback fish.” And we edited the sentence highlighted in this comment, now lines 356-358, to read “Focusing on comparisons between highland and lowland spring-fed populations allows us to test for effects of presumed divergence time on changes in antipredator behaviour, independent of the effect of sensory environment (because both have similar visibility, Figure S1).”.